

**Intended and Unintended Consequences of Atmospheric Methane Oxidation Enhancement**

Hannah M. Horowitz[1]

[1] Department of Civil and Environmental Engineering and Department of Climate, Meteorology, & Atmospheric Sciences, University of Illinois Urbana-Champaign, Urbana, Illinois, USA

*Correspondence to*: Hannah M. Horowitz (hmhorow@illinois.edu)

**Abstract.** Atmospheric oxidation enhancement (AOE) of methane via either tropospheric hydroxyl radicals (OH) or chlorine (Cl) radicals is being considered as a method to decrease greenhouse gas concentrations. The chemistry involved is coupled; is nonlinear; and affects air quality, other greenhouse gases, and ozone-depleting substances. Here I perform a suite of

experiments in a three-dimensional (3D) atmospheric chemistry model representing different OH- and Cl-based atmospheric oxidation enhancement methods, to estimate the effectiveness of each at decreasing greenhouse gases and the impacts on air quality and stratospheric ozone. I find that iron salt aerosol may not be effective at reducing methane on a global scale, depending on the reaction mechanism employed. More work is needed to understand the kinetics of chlorine release from iron salt aerosol and the potential for bromine co-release, which further decreases effectiveness. Hydrogen peroxide–based

approaches can decrease global methane, but the hydrogen peroxide emissions required may be too large to be feasible. I find that limiting emissions to daytime for hydrogen peroxide–based scenarios has negligible effects. All methods increase surface particulate matter (PM) pollution and in some regions lead to exceedances of annual air quality standards. Cl-based methods decrease ozone air pollution, but OH-based methods increase ozone air pollution in populated areas. While Cl-based methods can increase ozone-depleting substances, I predict minimal changes in stratospheric ozone after 1 year of

deployment. The overall impacts of atmospheric oxidation enhancement methods on climate and human health involve multiple competing factors.

## 1 Introduction

To achieve aggressive reductions in the greenhouse gas methane needed to reach climate goals, alternate strategies to emissions mitigation are being considered. These include processes to decrease the atmospheric lifetime of methane by

enhancing its main sinks, using oxidation by tropospheric OH (currently >90%) and tropospheric Cl (currently 1–5%) (e.g., Abernethy et al., 2023; Gorham et al., 2023; Li, Meidan, et al., 2023; Ming et al., 2022; Wang et al., 2022). There are interactions between Cl and OH; namely, increasing Cl decreases OH due to reductions in ozone (e.g., Horowitz et al., 2020; Li, Meidan, et al., 2023). In addition, Cl and OH can impact aerosol particles, other greenhouse gases, and ozone-depleting substances. Due to these interactions and the highly nonlinear chemistry involved, detailed atmospheric chemical

investigations are needed to understand the overall climate and pollution impacts of methods to enhance atmospheric



methane oxidation. To have the greatest impact on atmospheric methane, methods appropriate for ambient concentrations ($\leq$ 2 parts per million) are needed, but the technology is not yet available (Abernethy et al., 2023).

While commercial OH generators exist on a smaller scale (e.g., to remove volatile organic carbon pollution), it is less well understood how to scale up the artificial production of OH globally (e.g., Ming et al., 2022). Potential technologies could involve the release of hydrogen peroxide, downdraft energy towers, and/or artificial ultraviolet (UV) radiation (Tao et al., 2023; Wang et al., 2022). One specific method to release Cl atom is iron salt aerosol (e.g., Oeste et al., 2017), where the presence of both iron (III) and chloride enhances photolytic Cl atom release. Photolytic Cl atom release from artificial salts has been demonstrated in chamber experiments using salt pans and artificial sea salt aerosols generated from sodium chloride (NaCl) and seawater samples (Wittmer, Bleicher, Ofner, & Zetzsch, 2015; Wittmer, Bleicher, & Zetzsch, 2015). Recently, this process has been demonstrated to occur in the atmosphere in a study based on observations from Barbados through the mixing of natural iron-containing dust and sea salt aerosols and is thought to be due to the release of chlorine gas ($Cl_2$) followed by rapid photolysis (van Herpen et al., 2023). Gorham et al. (2023) present the state of the science for moving forward with intentional increase of this mechanism via iron salt aerosol. Meidan et al. (2024) find that the impact of iron varies depending on the region of application. Recent work has investigated the direct release of $Cl_2$ in a coupled chemistry–climate model and found that at least 90 Tg/yr $Cl_2$ is needed for Cl increases to outweigh decreases in OH with respect to overall methane loss, and at least 1,250 Tg/yr $Cl_2$ is needed to decrease the methane lifetime by 50% or more (Li, Meidan, et al., 2023).

Here I simulate in an atmospheric model a variety of methods of atmospheric oxidation enhancement (AOE) of methane via tropospheric OH and tropospheric Cl. I assess their impacts on methane, tropospheric chemistry, other greenhouse gases and ozone-depleting substances, and surface air quality.

## 2 Methods

I apply the global atmospheric chemical transport model GEOS-Chem (Section 2.1) with modifications (Section 2.2) to simulate atmospheric oxidation enhancement scenarios and their effects. The overall methodology of the study is outlined in Figure S1.

## 2.1 GEOS-Chem Model

Here I apply the 3D atmospheric chemical transport model GEOS-Chem version 13.2.1. (https://doi.org/10.5281/zenodo.5500717). As it is a chemical transport model, meteorological and climatic processes are not simulated directly. Instead, the model is driven by assimilated offline meteorological fields of the Modern-Era Retrospective Analysis for Research and Applications (Gelaro et al., 2017) from the National Aeronautics and Space Administration/Global Modeling and Assimilation Office. Simulations are performed at 4° latitude × 5° longitude horizontal resolution, with 72 vertical layers from the surface to the mesosphere (up to 80 km) and with active chemistry through the



stratosphere (up to 50 km). Simulations are performed for the arbitrary year 2019 following 1 year of initialization (year 2018).

Detailed coupled tropospheric halogen (chlorine, bromine, iodine) chemistry follows Wang et al. (2021) and
includes sea salt debromination and improvements to heterogeneous chemistry on polar stratospheric clouds (Eastham et al., 2014). Online stratospheric chemistry includes heterogeneous ozone depletion chemistry. Carbonaceous aerosol includes black carbon (Wang et al., 2014) and organic aerosol following the "simple" secondary organic aerosol (SOA) scheme (fixed-yield, direct, and irreversible formation) (Pai et al., 2020). Anthropogenic emissions follow the Community Emissions Data System (CEDS) v2 inventory originally developed for the Coupled Model Intercomparison Project Phase 6 (CMIP6)
(https://data.pnnl.gov/dataset/CEDS-4-21-21), including particulate iron emitted as a constant fraction of sulfur dioxide ($SO_2$). Sea salt aerosol emissions over the open ocean are wind- and sea surface temperature–dependent following Jaeglé et al. (2011), and in polar regions include blowing snow (Huang & Jaeglé, 2017). Dust emissions include natural (Fairlie et al., 2007) and anthropogenic dust from the anthropogenic fugitive, combustion, and industrial dust inventory (Philip et al., 2017). Biogenic volatile organic compound emissions are from MEGAN v2.1 (Guenther et al., 2012). All meteorologically
dependent emissions are calculated offline at the native model resolution ($0.5° × 0.625°$) (Weng et al., 2020) and appropriately scaled such that the total emissions are independent of model resolution (Lin et al., 2021). Wet deposition follows Amos et al. (2012) for gases and Liu et al. (2001) for aerosols, with snow and mixed precipitation scavenging from Wang et al. (2014). Dry deposition is a resistance-in-series approach (Wang et al., 1998) with aerosol dry deposition described in Zhang et al. (2001). Ozone deposition to the ocean via reaction with sea surface iodide follows Pound et al.
80 (2020).

Methane concentrations at the surface in GEOS-Chem are a fixed boundary condition based on monthly mean National Oceanic and Atmospheric Administration flask observations, after which methane advects and participates in chemistry (Murray, 2016). Hence, I use the methane feedback factor, which accounts for the feedback of methane on its own loss rate to estimate changes in steady-state methane concentrations. Impacts on steady-state methane concentrations
estimated using fixed boundary conditions along with the methane feedback factor are within 10% of impacts predicted from long (>40 years) simulations reaching equilibrium with fully responsive surface methane fluxes, while significantly reducing the computational cost (Khodayari et al., 2015). Fixed boundary conditions combined with the feedback factor also have been found to have a negligible impact on estimates of the global warming potential of hydrogen accounting for indirect methane feedbacks (Warwick et al., 2023). Here, I first calculate the methane lifetime following the same methods as in
Horowitz et al. (2020) from Holmes et al. (2013) and Holmes (2018). Briefly, the partial lifetimes of methane against tropospheric OH and Cl are calculated from the integrated 3D reaction rates. The total methane lifetime includes these two losses, stratospheric loss (assumed 120 years), and uptake to soil (assumed 150 years). Then, the change in steady-state methane concentration is estimated using the feedback factor $f$ following Holmes (2018):

$$\Delta[CH_4] = \left(\left(\frac{\tau_{exp}}{\tau_0}\right)^f - 1\right)[CH_4]_0 \qquad (1)$$





where $\tau_0$ and $\tau_{exp}$ are the lifetimes of methane in the standard and atmospheric oxidation enhancement experiments, respectively; $f = 1.34 \pm 0.06$ is the GEOS-Chem methane feedback factor on its loss rate (Holmes et al., 2013); and $[CH_4]_0 =$ 1,866.58 parts per billion (ppb) is the global annual mean surface methane concentration in year 2019 (https://gml.noaa.gov/ccgg/trends_ch4/).

**2.2 Model Experiments for Atmospheric Oxidation Enhancement**

A summary of simulations is presented in Figure S2. Scenarios are grouped between the dominant intended effect on increasing methane oxidation, either through reaction with OH (OH-based; Section 2.2.1) or Cl (Cl-based; Section 2.2.2). Under these umbrellas are experiments with different compounds emitted (OH or hydrogen peroxide for OH-based, and Cl$_2$ or particulate iron and/or chloride for Cl-based). All emissions are constant in time except for daytime-only tests (for simplicity, double the 24-hour emissions rate released between 6am and 6pm local time) for several of the hydrogen

peroxide emissions scenarios, as marked in Figure S2.

    For the remainder of the paper, I will focus on scenarios detailed in Table 1, which produced nonnegligible results. Direct OH release scenarios will be discussed briefly in the context of comparisons with the hydrogen peroxide emissions scenarios that led to comparable impacts. Details on the remaining simulations are included in the supplementary information; see Tables S1 and S2.




**Table 1: Details of focus scenarios in the current study.**

| | Emitted Species | Total Emissions (Tg/yr) | Emissions Location | Emissions Rate at Location of Emissions (kg/m$^2$/s) | Reaction Rate Coefficient |
|---|---|---|---|---|---|
| $H_2O_2$\_high | $H_2O_2$ | 1.61E7 | Globally at surface | 1.00E-6 | N/A |
| $H_2O_2$\_mid | $H_2O_2$ | 7.4E4 | Globally at surface | 4.6E-9 | N/A |
| $H_2O_2$\_low | $H_2O_2$ | 1,250 | Globally at surface | 7.7E-11 | N/A |
| $Cl_2$\_ocean | $Cl_2$ | 1,250 | Oceans | 1.10E-10 | N/A |
| $Cl_2$\_BrCl\_$Br_2$ | $Cl_2$ | 1,193 | Oceans | 1.045E-10 | N/A |
| | BrCl | 187 | | 1.137E-11 | |
| | $Br_2$ | 129 | | 1.64E-11 | |
| Iron | pFe | 565 | Oceans | 4.97E-11 | $\frac{d[Cl_2]}{dt}= \alpha j_{NO2}\,[Fe^{3+}][Cl^-]S$ |
| Iron\_Max | pFe | 1,250 | Oceans | 1.10E-10 | |
| Chloride | Accumulation mode chloride | 1,250 | Oceans | 1.10E-10 | |
| Iron\_Chloride | Accumulation mode chloride | 1,250 | Oceans | 1.10E-10 | |
| | pFe | 565 | | 4.97E-11 | |

### 2.2.1 Hydroxyl Radical

#### 2.2.1.1 Hydrogen peroxide emission

Hydrogen peroxide photolysis produces two OH radicals, and the emission of hydrogen peroxide has been proposed as a possible method to enhance atmospheric methane oxidation via OH (Wang et al., 2022). For example, there is a patent to deploy hydrogen peroxide towers for this purpose (Bell, 2023). The concurrent production of hydrogen peroxide and release of OH via Fenton-like catalytic processes has also been suggested as a potential technology (Wang et al., 2022).

     In this study, I tested four scenarios to investigate the impact if the current global demand of 4.1 Tg/yr $H_2O_2$
(Research and Markets, 2023) was released additionally (see Figure S2): globally at the surface, only at major point sources of oil and gas emissions determined from the CEDS anthropogenic emissions inventory (0.2% of Earth's area at the 4° × 5°



model resolution) at the surface, 600 m (stack height proposed by the Bell, 2023 patent), and 600 m emitted only during daytime to maximize photolysis. All 4.1 Tg/yr scenarios produced negligible effects. The change to 600 m stack height as well as from 24-hour to daytime-only did slightly increase the impact on methane but still resulted in <0.5 ppb (≤0.03%)

decrease in methane. As the spatial resolution is low (4° × 5° or 400–500 km at midlatitudes), this current model study may not address fully whether this method is viable for individual, large point sources of methane. Given the negligible results found here, these scenarios were not investigated further in this study.

For the remainder of the paper, I will focus on three global hydrogen peroxide emissions scenarios to span a range of impacts on methane and feasibility ($H_2O_2\_high$, $H_2O_2\_mid$, and $H_2O_2\_low$; see Table 1). All scenarios were emitted at the

surface (0 to 12.3 m altitude in GEOS-Chem). In the 4.1 Tg/yr point source tests, impacts increased when the hydrogen peroxide was emitted at 600 m with respect to the surface. It is thus possible that the impacts on methane by these global scenarios could be larger if emitted at 600 m; this would require further investigation. I also tested $H_2O_2\_mid$ and $H_2O_2\_low$ with double the 24-hour emissions rate only released during 6am–6pm local time. Daytime-only emissions did increase the impact on methane, with a larger relative effect for the smaller hydrogen peroxide emissions, but the absolute values of

methane lifetime and steady-state methane concentrations are within ±1% for the corresponding daytime-only and 24-hour emissions scenarios. Thus, I will focus on results from the 24-hour emissions for the remainder of the paper.

### 2.2.1.2 OH chemical production

Not all hydrogen peroxide is immediately photolyzed to produce OH and may undergo alternate reactions. Thus, to

examine the direct effects of OH release in the absence of other chemical changes, I introduced a dummy reaction in GEOS-Chem to produce OH from $O_2$, a species with fixed concentrations in GEOS-Chem (0.2095 mol/mol), from the surface up to 1 km altitude globally (see Table S2 in the supplementary information). OH cannot be advected in GEOS-Chem due to its short lifetime and thus it cannot be emitted directly. I tested two rates, OH_mid and OH_high; OH_high's rate of OH emission is two times higher (see Table S2) to result in similar changes as the hydrogen peroxide simulations. In reality, OH

could be released directly through methods incorporating artificial UV radiation (e.g., Ming et al., 2022) or downdraft energy towers that generate electricity from seawater and sunlight and produce additional OH from ozone due to the water vapor introduced in dry regions (Tao et al., 2023; Wang et al., 2022); these would likely be point sources that are not feasible in the current modeling framework. Given this and the fixed methane boundary conditions, the direct OH results will only be discussed in reference to how air quality impacts change when OH versus hydrogen peroxide is emitted at comparable

impacts on methane.

### 2.2.2 Chlorine

### 2.2.2.1 Direct $Cl_2$ emission

In the $Cl_2$ simulation, $Cl_2$ is directly emitted across the global oceans at the surface, a total of 1,250 Tg/yr, which is the midrange scenario for methane removal in Li, Meidan et al. (2023). This is approximately 20 times higher than the





current total tropospheric source of gas-phase inorganic chlorine in GEOS-Chem (Wang et al., 2021). Emissions of $Cl_2$ at the surface are more likely to be economically and technologically feasible and will limit the impact of additional chlorine on stratospheric ozone (Li, Meidan, et al., 2023). $Cl_2$ will then photolyze to produce two Cl atoms, of which approximately 20% will react with ozone instead of methane over the oceans (Li, Meidan, et al., 2023). Table S3 compares the modeling setup of the current study and Li, Meidan et al. (2023), including differences in resolution, year, and halogen chemistry.

**2.2.2.2 Bromine contamination**

It is not possible to remove 100% of bromide from pure chlorine salts. This could lead to reactive bromine release that would decrease OH in the atmosphere through the destruction of ozone, the primary $HO_x$ ($HO_x = OH + HO_2$) source (e.g., Horowitz et al., 2020), while unlike Cl it does not also oxidize methane. Here I create a model experiment ($Cl_2\_BrCl\_Br_2$) including direct emission of the bromine species $Br_2$ and BrCl. Wittmer, Bleicher, and Zetzsch (2015)

measured the Br versus Cl production rate from artificial sea salt created from a variety of iron-containing artificial seawater or NaCl stock solutions, with the ratio of Br/Cl produced by mass ranging from 0 to a factor of 2.5. Br atom was below the detection limit for the NaCl-based solutions (Wittmer, Bleicher, & Zetzsch, 2015), which may better resemble artificially engineered iron salt aerosol. However, previous work with salt pans found that even when Br atom was below the detection limit, bromide impurities in "pure" NaCl ($\leq 0.01\%$ Br$^-$) could lead to Cl atom release due to BrCl formation and photolysis,

which would also release Br atoms in equal quantities (Wittmer, Bleicher, Ofner, & Zetzsch, 2015). Here I assume that of the total desired chlorine release (1,250 Tg/yr as in the $Cl_2$-ocean simulation), 20% of that by mass of bromine is released in equal parts $Br_2$ and BrCl. More laboratory studies are needed to understand the potential bromine release from engineered iron salt aerosol.

**2.2.2.3 Iron salt aerosol**

Here I implement a parameterization of photolytic $Cl_2$ release from iron-enriched salt aerosol developed by Chen et al. (2024). The production rate of $Cl_2$ ($d[Cl_2]/dt$: molec cm$^{-3}$ s$^{-1}$) is a function of the nitrogen dioxide ($NO_2$) photolysis frequency ($j_{NO2}$: s$^{-1}$), accumulation mode aerosol iron (III) concentration ([$Fe^{3+}$]: mol l$^{-1}$), accumulation mode aerosol chloride concentration ([$Cl^-$]: mol l$^{-1}$), and aerosol surface area concentration (S: μm$^2$ cm$^{-3}$), and is scaled by the factor α (=1.4 × 10$^5$ molec μm$^{-2}$ M$^{-2}$) based on experimental results from Wittmer, Bleicher, and Zetzsch (2015) accounting for the

volume of the chamber:

$$d[Cl_2]/dt = \alpha j_{NO2}[Fe^{3+}][Cl^-]S \qquad (2)$$

This reaction occurs on accumulation mode chloride aerosol, which in GEOS-Chem is defined as $\leq 0.5$ μm in diameter.

In the standard GEOS-Chem model, Fe(III) concentrations are estimated from natural dust and anthropogenic particulate Fe for the Fe(III)-catalyzed $SO_2$ oxidation chemistry. This assumes that total Fe content from dust is 3.5% of total

mass (Taylor & McLennan [1985], consistent with Trapp et al. [2010]), dust iron solubility is 1% (Alexander et al., 2009) while anthropogenic particulate Fe is more soluble at 10% (Shao et al., 2019), and Fe(III) is 10% of total dissolved iron in the daytime (Moffet et al., 2012) and 90% at night. These assumptions are not sufficient to match observations in polluted urban environments in China (Chen et al., 2024). Model evaluation against observations suggests four times higher solubility



for both dust and anthropogenic Fe (4% and 40%, respectively) and 67% of total dissolved iron as Fe(III) (Chen et al., 2024).
Observed Fe solubility in aerosols over the oceans is highly variable (0.19% to 47.8%) and is influenced by the initial source of the iron (e.g., dust vs. combustion), other chemical components in the aerosol, pH, and relative humidity (Shi et al., 2022). For the iron salt aerosol simulations, I use the Chen et al. (2024) representation of solubility and Fe(III) speciation as a maximum rate of $Cl_2$ production through this particular parameterization. van Herpen et al. (2023) also parameterized the production of $Cl_2$ from natural mineral dust–sea spray aerosols. Major differences between Chen et al. (2024) and van
Herpen et al. (2023) include the size of the particles on which this process occurs, the percentage of photoactive iron, and that the Chen et al. (2024) study is based primarily on the Wittmer, Bleicher, and Zetzsch (2015) chamber experiments while van Herpen et al. (2023) follow the Fe(II)–Fe(III) cycling kinetics from Zhu et al. (1993). Differences in the model frameworks are summarized in Table S4.

First I perform GEOS-Chem simulations with the additional reaction in the absence of any additional emissions to
act as a reference point for the atmospheric oxidation enhancement experiments. Then, to release additional $Cl_2$ from iron salt aerosol, accumulation mode aerosol chloride and particulate iron (pFe) are released over the oceans at the surface in four scenarios to assess the driving factors and compare against the direct $Cl_2$ scenario. The highest iron addition tested in Wittmer, Bleicher, and Zetzsch (2015) was 13 mol $Cl^-$/mol $Fe^{3+}$. Given the assumed solubility of anthropogenic pFe of 40% and Fe(III) speciation fraction of 67% in my GEOS-Chem simulations, a ratio of 3.484 mol $Cl^-$/mol pFe emitted would lead
to Fe(III) ratios at most comparable to the highest iron addition experiments in Wittmer, Bleicher, and Zetzsch (2015). For chloride emissions of 1,250 Tg, this is 565 Tg/yr pFe. The experiments here include iron emissions alone (Iron), chloride emissions alone (Chloride), a combination of both iron and chloride (Iron_Chloride), and iron alone emitted at 1,250 Tg/yr (Iron_Max) as a test case (see Table 1 for details). Iron_Max does not lead to significantly more methane loss than the Iron scenario; hence, it will only be discussed in the context of methane and not with respect to tropospheric chemistry and air
quality.

## 3 Results

### 3.1 Impacts on Tropospheric Chemistry

#### 3.1.1 Major Tropospheric Oxidants and Reactive Halogens

Table 2 presents the tropospheric burdens of major oxidants, reactive halogen families, and carbon monoxide (CO)
for the standard GEOS-Chem simulation and the standard simulation plus the Chen et al. (2024) parameterization, followed by relative percent changes in these burdens for the atmospheric oxidation enhancement model experiments described in Table 1.

**Table 2. Percent change in simulated annual mean tropospheric burdens of selected species and chemical families.**



|  | Br$_y$ | Cl$_y$ | I$_y$ | O$_3$ | OH | Cl | NO$_x$ | CO |
|---|---|---|---|---|---|---|---|---|
| Standard | 20 Gg | 241 Gg | 12 Gg | 338 Tg | 215 Mg | 318 kg | 359 Gg | 349 Tg |
| Standard+Chen | 20 Gg | 241 Gg | 12 Gg | 338 Tg | 215 Mg | 333 kg | 359 Gg | 349 Tg |
| H2O2_high | -11.1 | 36.7 | -77.5 | -38.5 | 165.3 | 396.4 | 8.3 | -46.3 |
| H2O2_mid | 48.2 | 12.3 | -20.6 | -6.3 | 30 | 75.2 | -17.2 | -23 |
| H2O2_low | 11.0 | 0.9 | -1.4 | -0.8 | 2.3 | 6.5 | -4.1 | -2.8 |
| Cl2 | -6.7 | 1738 | -42.7 | -24.4 | -27.6 | 2185.2 | -18.7 | 52.1 |
| Cl2_BrCl_Br2 | 2567 | 1689 | -75.4 | -67.7 | -47.6 | 1839.8 | -31.1 | 100.1 |
| Iron* | 7.1 | 12.1 | -5.5 | -3.5 | -2.2 | 179.3 | -2.6 | 3.3 |
| Chloride* | 45.5 | 1114.6 | -8.6 | -5.5 | -3.9 | 180.9 | -3.5 | 4.2 |
| Iron_Chloride* | 42.7 | 1139 | -17.3 | -10.7 | -8.6 | 680.8 | -9.3 | 13.5 |

Note: Percent change in annual mean tropospheric burdens for model experiments (described in Table 1) are relative to the standard version (* = relative to standard + Chen et al., 2024). Br$_y$, Cl$_y$, and I$_y$ follow definitions in Wang et al. (2021); NO$_x$ =NO + NO$_2$. See also Section 2.1.

Previous work on an intermodel comparison of atmospheric chemistry and climate models from the Atmospheric Chemistry and Climate Model Intercomparison Project found that changes in global mean OH between models and within a
given model are more a function of the relative loss of reactive nitrogen versus reactive carbon than the emissions of reactive nitrogen versus carbon, as it matters how much of the loss of these species is due to OH versus other processes (Murray et al., 2021). The loss of reactive carbon is a function of a given model's chemical mechanism and structure (Murray et al., 2021). The GEOS-Chem model version used in this study (13.2.1) is several generations ahead of that used in Murray et al. (2021) (version 9-01-03), with a number of changes in reactive nitrogen chemistry including aerosol uptake and recycling of
isoprene nitrates (e.g., Fisher et al., 2016, 2018; Holmes et al., 2019; McDuffie et al., 2018) that have been shown to impact the tropospheric OH burden by up to 12%. The representation of nitrogen oxides (NO$_x$) loss thus can lead to uncertainty in the simulated OH burden, but it is not clear if this would be a consistent bias across all model experiments in the current study and thus negligible when considering relative changes.

As expected, the hydrogen peroxide experiments all increase tropospheric OH due to photolysis of the additional
hydrogen peroxide. OH does not increase proportionally to the increase in hydrogen peroxide emissions across experiments and indicates a reduction in the effectiveness of additional emissions at higher levels of hydrogen peroxide. A >200-fold increase in hydrogen peroxide emissions from H$_2$O$_2$_mid to H$_2$O$_2$_high leads to a 5.5-fold increase in OH; a 60-fold increase in hydrogen peroxide emissions from H$_2$O$_2$_low to H$_2$O$_2$_mid leads to a 13-fold increase in OH (see Table 2). The increases in OH drive reductions in tropospheric CO seen in these experiments, as OH is the main oxidant of CO. The reductions in
CO are nearly proportional to the OH increases in the H$_2$O$_2$_low and H$_2$O$_2$_mid scenarios; however, in the extreme H$_2$O$_2$_high scenario, the CO decrease is much lower than expected from the OH increase (see Table 2). This is likely due to



the much larger increase in Cl atom in the H$_2$O$_2$_high experiment, leading to additional production of CO from Cl reactions with formaldehyde and organochlorines. In all of the hydrogen peroxide experiments, Cl atom increases. Increased HO$_x$ (= OH + HO$_2$, which cycle rapidly between each other) leads to increased release of chlorine from sea salt aerosol, increasing

the total Cl$_y$ burden. In addition, the partitioning of gas-phase Cl$_y$ shifts away from the longer-lived reservoir species hydrogen chloride (HCl) toward chlorine hydroxide (HOCl), Cl$_2$, and Cl, such that the relative increase in Cl atom is greater than the relative increase in total Cl$_y$ in each experiment. This is due to the complex interplay of cross-reactions between fast-cycling HO$_x$ and chlorine radicals as well as across their longer-lived reservoir species.

        Cl-based experiments decrease tropospheric OH due to the destruction of ozone by the additional Cl, as ozone

provides the main source of tropospheric OH. Tropospheric CO increases in the chlorine-based experiments, partially due to the decrease in OH as well as the additional production of CO from reactions of formaldehyde and organochlorines with Cl atom. The Cl$_2$, Chloride, and Iron_Chloride experiments have the same amount of total chlorine emissions (see Table 1) but vastly different effectiveness at increasing the Cl atom concentration. The Chloride and Iron_Chloride experiments, where particulate accumulation mode chloride was emitted, have 65% as much of an increase in total gas-phase Cl$_y$ burden relative

to the Cl$_2$ experiment where gas-phase Cl$_2$ was emitted (see Table 2). However, only 8% (Chloride only) to 31% (Iron_Chloride) of the increase in Cl atom seen in the Cl$_2$ experiment is realized. This is because the vast majority of the increase in Cl$_y$ burden in these experiments is due to HCl, shifting Cl$_y$ away from more reactive species. The release of Cl$_2$ is iron-limited, as the addition of particulate iron emissions with the same amount of chloride (from Chloride to Iron_Chloride) leads to a nine-fold increase in Cl$_2$ and a three-fold increase in Cl.

260        In the Cl$_2$ and Cl$_2$_BrCl_Br$_2$ experiments, Cl$_y$ partitioning is shifted away from HCl to Cl$_2$ and Cl. Thus, the relative increase in Cl atom is even greater than the relative increase in total tropospheric Cl$_y$ (see Table 2). In Li, Meidan et al. (2023), 1,250 Tg/yr Cl$_2$ lead to a 30-fold increase in the tropospheric Cl burden; these emissions in our shorter-term modeling study led to a 23-fold increase in the Cl$_2$-only case and a 19-fold increase in the Cl$_2$_BrCl_Br$_2$ case. This may be due to differences in the reactive halogen reactions and rate coefficients between the model used here versus in Li, Meidan et

al. (2023) and the timescale of the study (see Table S3).

        All experiments decrease the tropospheric ozone burden (see Table 2), with implications for ozone radiative forcing. All experiments have increased Cl atom concentrations, which can destroy ozone. H$_2$O$_2$_low, H$_2$O$_2$_mid, and all chlorine experiments except Cl$_2$ also have increased Br$_y$ burdens, which can lead to additional ozone loss via reactive bromine. In the H$_2$O$_2$_low, H$_2$O$_2$_mid, and chlorine experiments, part of the change in ozone is also due to reductions in NO$_x$

that lead to reduced ozone production. The increases in Br$_y$ and decreases in NO$_x$, which became larger as hydrogen peroxide emissions increase from H$_2$O$_2$_low to H$_2$O$_2$_mid, flip sign once hydrogen peroxide emissions become particularly extreme in the H$_2$O$_2$_high scenario. Thus in the H$_2$O$_2$_high case, the ozone loss is likely dominated by the large increase in Cl atom (see Table 2).

        As seen in Horowitz et al. (2020), the decreases in ozone drive decreases in tropospheric I$_y$ in all experiments, as

ozone reactions with sea surface iodide are the dominant source of tropospheric I$_y$ (Wang et al., 2021). The greatest impacts





on ozone are in the Cl$_2$_BrCl_Br$_2$ experiment, which included bromine release to represent bromide contamination (see Table 1). The 20% additional mass released as bromine (Br$_2$ and BrCl, which both photolyze to release Br atom) led to a much larger than 20% additional reduction in ozone over the Cl$_2$-only experiment (factor of 2.78 higher) and hence OH (factor of 1.72 higher) (see Table 2). While 20% is likely an overestimate of the bromide content of engineered aerosol, it is

well within the range reported from natural seawater salt experiments catalyzed by iron (Wittmer, Bleicher, & Zetzsch, 2015) and suggests that even a small perturbation in bromine could have a much larger impact on OH.

The OH_mid and OH_high scenarios lead to increases in OH that are larger than the H$_2$O$_2$_mid experiment but smaller than H$_2$O$_2$_high (see Table S5). The behaviors of the other tropospheric species in Table 2 are all similar to the hydrogen peroxide experiments, except for CO (decrease is less than expected given OH increase) and Cl (hydrogen

peroxide experiments have a relative increase in Cl that exceeds that of the relative increase in OH; for OH experiments it is less). The OH experiments' results for CO are less reliable as the OH increase is limited to the lower 1 km where it is released. (OH is not transported in the model; hydrogen peroxide is.) In this region, methane levels remain high due to the surface boundary condition and long lifetime against reaction with OH; as CO is reacted away by the additional OH, its loss rate slows down due to its shorter lifetime and fully responsive concentrations at the surface.

**3.1.2 Impacts on Methane**

Figure 1 (top panel) summarizes the impacts on the overall methane lifetime, the impacts on the methane lifetime with respect to tropospheric OH, and the percent of Cl atom contributing to total chemical loss in the troposphere for each of the experiments in Table 1 and the standard model. Figure 1 (bottom panel) shows the resulting relative change in steady-state methane concentrations estimated with the methods described in Section 2.1.



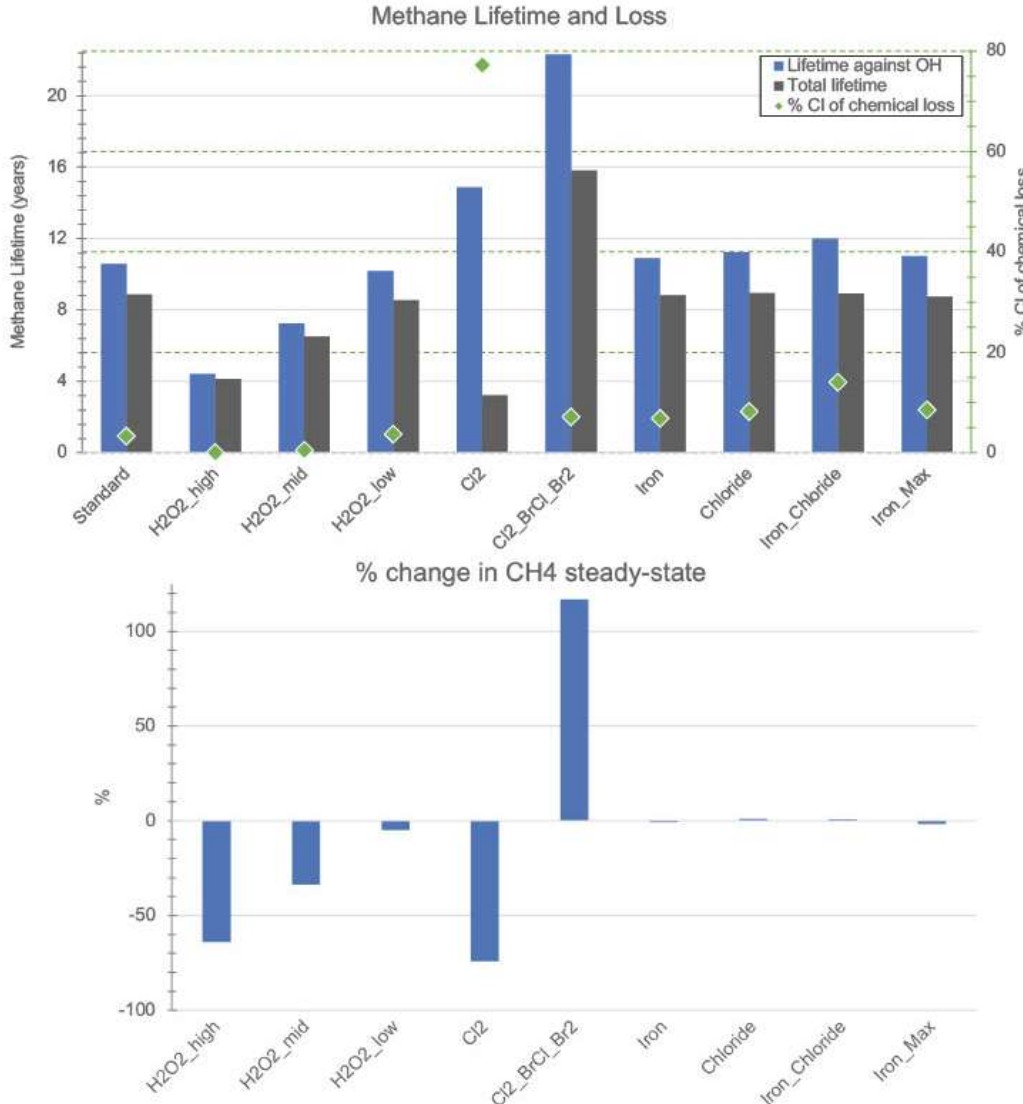


**Figure 1: Top panel: total methane lifetime (gray bars) and methane lifetime against oxidation by OH (blue bars), overlaid by the percent of chemical loss contributed by Cl atom (green diamonds, right-hand y-axis). Bottom panel: estimated relative percent change in steady-state methane concentration.**



Studies suggest that the present role of chlorine in tropospheric methane oxidation is 1–5% based on constraints from isotopic observations (Allan et al., 2007; Gromov et al., 2018; Platt et al., 2004) and atmospheric modeling studies focused on chlorine chemistry (e.g., Hossaini et al., 2016; Wang et al., 2019). The standard version of GEOS-Chem used here is consistent with these constraints at 3.3%, which gives confidence to the simulation of atomic chlorine and its reaction with methane.

Although the $H_2O_2$_high experiment had the largest increase in tropospheric OH, the chemical loss through the Cl pathway is shut down (Figure 1), leading to a smaller impact on the methane lifetime relative to the comparable OH scenarios (not shown). Significant methane reductions for the hydrogen peroxide scenarios (>5%) require more than 1,250 Tg/yr emissions ($H_2O_2$_low).

    For the same annual emissions of $Cl_2$ (1,250 Tg/yr) as in Li, Meidan et al. (2023), I find similarly large decreases in
the methane lifetime (here ~67% vs. 50%) and methane concentrations (here –74% at steady-state vs. –45% in year 2050) despite different modeling frameworks. While the tropospheric burden of Cl increases more in Li, Meidan et al. (2023), as described in the previous section, I find a larger percent of chemical loss due to tropospheric chlorine (77% vs. 60%). This may be due to differences in the vertical distribution of Cl as well as the meteorology between the two models, as the Cl + $CH_4$ reaction is temperature and air-density dependent. The vertical distribution of Cl in the two models is not only impacted
by the halogen chemical mechanism (see Table S3); in GEOS-Chem, an improved representation of entrainment-limited uptake in clouds and ice cloud particle properties for cloud heterogeneous chemistry leads to significant increases in reactive chlorine, particularly in the upper troposphere, due to changes in HCl-$ClO_x$ (chlorine oxides) cycling (Holmes et al., 2019). Other differences that could impact the results are the increase in methane emissions in Li, Meidan et al. (2023) following the representative concentration pathway 8.5 scenario, and the shorter simulation period in our study (see Table S3).
Bromine contamination of 20% more than reverses the gains seen in the $Cl_2$ case, as the $Cl_2$_BrCl_$Br_2$ case has an increase in steady-state methane concentrations of 117% (see Figure 1).

    van Herpen et al. (2023) found that including their parameterization of $Cl_2$ release from mineral dust–sea spray aerosol (without any additional emissions) led to an increase in methane loss via the Cl atom of ~20%, although the overall global methane loss was decreased slightly due to a reduction in methane loss via reaction with OH. Here I find an increase
in the tropospheric burden of Cl atom of ~5% (see Table 2) when the Chen et al. (2024) parameterization is included (without any additional emissions), but this only translates to an increase of 0.2% in the methane loss via Cl atom. While van Herpen et al. (2023) do not report global changes in chlorine burdens, they find that total inorganic chlorine production is increased by 41%. This led to a change in –0.7% in the tropospheric ozone burden after including $Cl_2$ release from natural iron salt aerosol (van Herpen et al., 2023), while I find a change of –0.07% in the tropospheric ozone burden when I include
the Chen et al. (2024) parameterization. Thus, it is likely that the two parameterizations (see Table S4) result in $Cl_2$ production rates that are up to 10 times higher in van Herpen et al. (2023).

    With the parametrization used in this study, iron salt aerosol cannot produce enough chlorine to overcome the decrease in methane loss via the OH channel. Changes in steady-state methane for iron salt scenarios are < ± 2% (see Figure



1). While the Iron experiment led to a very small decrease in steady-state methane (–0.8%) from a factor of 2.8 increase in Cl atom, the Iron_Chloride experiment led to a small increase in steady-state methane (+0.7%) in spite of a larger 7.8-fold increase in Cl atom burden due to a much larger reduction in OH (see Table 2). For the same 2.8-fold increase in Cl atom burden, emitting chloride (Chloride) instead of iron (Iron) similarly led to a larger decrease in OH and hence net increase in methane (+0.8%). Li, Meidan et al. (2023) found that a 2.8-fold increase in Cl burden (from 88 Tg/yr gas-phase $Cl_2$ emission) was insufficient to decrease methane, while a 7.9-fold increase in Cl burden (from 313 Tg/yr gas-phase $Cl_2$ emissions) overcame the OH competition and led to a decrease in methane concentrations by about 6% after 10 years (Li, Meidan, et al., 2023). This suggests that the threshold of additional chlorine needed to overcome the OH limitation depends on what is emitted. Here I find that emitting sea salt chloride instead of or along with particulate iron worsens methane outcomes.

The iron emissions needed to decrease methane are also uncertain and sensitive to the mechanism and rate of chlorine release from iron salt aerosol. The Iron and Iron_Chloride scenarios presented here have 565 Tg/yr iron emissions (see Table 1). In a Community Earth System Model (CESM) study using the van Herpen et al. (2023) parameterization for chlorine release, additional iron emissions of 200 Tg/yr (which increased $Cl_2$ production by ~850 to 930 Tg/yr depending on the region of emission) lead to a ~20% decrease in global methane concentrations after 10 years (Meidan et al., 2024). This study found that a threshold of at least 6 Tg/yr of iron was needed in the most idealized setup to avoid increasing methane, though larger emissions of >60 Tg/yr of iron may be needed for significant methane reductions.

Here, the methane loss in the Iron_Chloride scenario decreases in the tropical and southern hemisphere's free troposphere, with increases in methane loss rates limited mostly to the surface and some regions of the upper troposphere. There may also be a reduction in $Cl_2$ production from chloride via other reactions with OH and chlorine nitrate ($ClNO_3$) due to reductions in tropospheric OH and $NO_x$; thus, the representation of competing heterogeneous halogen chemical reactions may play a role in the predicted effectiveness of iron salt aerosol. Additional laboratory experiments and field observations are needed to constrain this process in natural and engineered aerosol mixtures.

Overall, OH-based scenarios and gas-phase emission of $Cl_2$ lead to significant decreases in steady-state methane but require extremely large emissions. $Cl_2$ is much more effective at reducing methane (–77%) than hydrogen peroxide (–5%) for the same level of emission (1,250 Tg/yr). There are uncertainties in the exact impact on methane of a given amount of $Cl_2$ emission (e.g., in this study vs. Li, Meidan, et al., 2023) due to a variety of factors including the representation of complex reactive halogen chemistry. Despite this, the large methane lifetime and methane concentration reductions from 1,250 Tg/yr of gas-phase $Cl_2$ emissions presented here are qualitatively similar to those of Li, Meidan et al. (2023). In contrast, by using a different chlorine release mechanism from iron salt aerosol from Chen et al. (2024), I find no significant reductions in methane for even larger iron emissions than those used in Meidan et al. (2024). The mechanism of $Cl_2$ emission, whether this would also release bromine, and how fast the release from iron salt aerosol is will affect whether this method would decrease or increase steady-state methane.



### 3.1.3 Impacts on Other Climate Forcers and Ozone-Depleting Substances

GEOS-Chem simulates additional greenhouse gases and ozone-depleting substances including nitrous oxide ($N_2O$), halons (three species), hydrochlorofluorocarbons (HCFCs; four species), chlorofluorocarbons (CFCs; five species), and other

halomethanes (12 species, e.g., chloromethane, methyl bromide, chloroform). I examined the changes in tropospheric and stratospheric burdens of each. As our simulation is performed for only 1 year following 1 year of initialization, stratospheric changes do not represent the long-term impact of atmospheric oxidation enhancement of methane due to the long lifetime of air in the stratosphere. At the same time, stratospheric transport and chemistry in GEOS-Chem previously have been evaluated to perform well for stratospheric applications. Earlier versions of GEOS-Chem simulations conducted at the same

horizontal and vertical resolution as the current study have shown that age of air at 20 km altitude for all latitudes is within ±6 months of that observed (Eastham et al., 2022), and performs well against in-situ and satellite observations of stratospheric ozone depletion, tracer-tracer correlations, and lifetime of long-lived gases in the stratosphere (Eastham et al., 2014).

Details of changes in tropospheric and stratospheric burdens (from the temporally and spatially varying tropopause

up to 50 km) for selected species with significant impacts from the AOE experiments are shown in Tables S6 and S7. For CFCs, halons, $N_2O$, and most halomethanes, impacts are negligible. Tropospheric burdens of HCFC-123 decrease in the hydrogen peroxide–based scenarios and increase in the Cl-based scenarios. Bromide contamination of 20% in the $Cl_2\_BrCl\_Br_2$ experiment more than doubles the increase in HCFC-123 relative to the $Cl_2$-only experiment. HCFC-123 has a 100-year global warming potential of 90.4 (Smith et al., 2021) and is a class-II ozone-depleting substance. Changes in the

stratospheric abundance of HCFC-123 are, however, negligible. Of the remaining halomethanes, bromoform ($CHBr_3$) and dibromomethane ($CH_2Br_2$) have the largest changes in the troposphere and stratosphere, with decreases in the hydrogen peroxide–based experiments and increases in the Cl-based experiments. Dichloromethane and chloroform follow the same pattern in the troposphere and stratosphere, but changes are smaller overall. For dibromoethane, dichloromethane, and chloroform, the addition of bromine in the $Cl_2\_BrCl\_Br_2$ experiment again leads to larger increases relative to $Cl_2$ only.

These short-lived gases could contribute to ozone depletion in the stratosphere and are not regulated by the Montreal Protocol (e.g., Hossaini et al., 2017). Iodine-containing halomethane tropospheric and stratospheric burdens decrease across all experiments due to the reduction in overall $I_y$ (Section 3.1.1), with the largest decreases seen for methyl iodide ($CH_3I$), which is not thought to contribute to stratospheric ozone loss (Zhang et al., 2020). Li, Meidan et al. (2023) found increases in stratospheric Cl and thus stratospheric ozone loss from the release of $Cl_2$ for AOE of methane. (In their study, the

stratosphere was up to 40 km.) In this study, stratospheric chlorine up to 50 km is negligibly affected in the hydrogen peroxide–based experiments with a decrease of –1.5 to –3.1% in the Cl-based experiments. This difference may reflect the short duration of the simulations performed here (1 year vs. 30 years). Overall, in this study, stratospheric ozone changes negligibly in all experiments except in the $Cl_2\_BrCl\_Br_2$ scenario, where the stratospheric ozone burden decreases by –7.8%.



Across all experiments except Chloride, tropospheric inorganic aerosol increases (see Table S6) due to the increase
in tropospheric sulfate and ammonium, partially offset by decreases in tropospheric nitrate burdens. This is likely due to the
HOCl/HOBr+S(IV) pathways that produce sulfate (Wang et al., 2021), as $Br_y$ and $Cl_y$ both increase in nearly all
experiments. This is a departure from Li, Meidan et al. (2023), who found decreases in sulfate of ~10% from 1,250 Tg/yr $Cl_2$
emissions. This difference may reflect the coupled halogen-sulfate chemistry in GEOS-Chem (Chen et al., 2017; Wang et al.,
2021) that does not appear to be included in the CESM chemical mechanism (see Table S3). For the hydrogen peroxide
experiments, increased hydrogen peroxide and OH will also increase the oxidation of $SO_2$ to sulfate in the aqueous and gas
phases, respectively. For the Iron and Iron_Chloride experiments, sulfate may also increase due to the Fe(III)-catalyzed
pathway (Alexander et al., 2009). Partitioning of ammonium and nitrate aerosol is determined by the thermodynamic model
ISORROPIA-2 in GEOS-Chem (see Section 2.1). Thus, ammonium likely concurrently increases to neutralize the excess
sulfate in all experiments. For the Chloride experiment, decreases in nitrate and sulfate are nearly exactly offset by increases
in ammonium. Nitrate is likely reduced due to reductions in $NO_x$ in most experiments (see Table 2) and is influenced by the
change in gas-particle partitioning due to increased sulfate production. Increases in the tropospheric inorganic aerosol burden
could lead to an additional negative radiative forcing. Changes in these aerosols at the surface and the associated air quality
implications are examined in Section 3.1.4.

In the stratosphere, total inorganic aerosol increases in the $Cl_2$ and $Cl_2\_BrCl\_Br_2$ scenarios due to increased sulfate
(see Table S7). Stratospheric total inorganic aerosol in the hydrogen peroxide experiments decreases by ~2%. This does not
impact stratospheric ozone.

### 3.1.4 Impacts on Surface Air Quality

Surface $PM_{2.5}$, CO, ozone, and $NO_2$ are air pollutants impacted by the atmospheric oxidation enhancement
experiments. These largely follow the changes in tropospheric burdens (see Tables 2 and S6). Surface $NO_2$, however, was
negligibly impacted by all experiments; most of the changes in tropospheric $NO_x$ occurred in the upper troposphere (see
Figure S3). The change in global annual mean surface $PM_{2.5}$, CO, and ozone is shown in Figure S4 across the different
scenarios, with spatial variations in annual mean surface $PM_{2.5}$ and ozone shown in Figures 3 and 4, respectively.

In the mean across all scenarios, there are surface ozone co-benefits (Figure S4). Surface $PM_{2.5}$ increases in all
scenarios except in the Chloride case, where in the absence of iron to mediate the $Cl_2$ release, the additional chloride leads to
decreased sulfate and nitrate. Interestingly, $H_2O_2\_mid$ has worse air quality impacts than $H_2O_2\_high$ despite having less
emissions; $PM_{2.5}$ increases more, and CO and ozone decrease less. There are larger impacts on $PM_{2.5}$ from the hydrogen
peroxide and OH methods (up to 14–19 $\mu g/m^3$) than the $Cl_2$ and $Cl_2\_BrCl\_Br_2$ methods (up to 6–7 $\mu g/m^3$). Figure 2 shows
the absolute change in $PM_{2.5}$ spatially. Gray boxes highlight areas already in exceedance of the U.S. Environmental
Protection Agency's (EPA's) annual mean $PM_{2.5}$ primary standard of 9 $\mu g\ m^{-3}$, which remain in exceedance when the
experiment is applied. Black boxes highlight areas where the experiment to increase methane oxidation brought a region





from <9 µg m$^{-3}$ to an exceedance of 9 µg m$^{-3}$. Blue boxes show where the experiment led to improved PM$_{2.5}$ from an exceedance to < 9 µg m$^{-3}$.

Although the emissions for Cl$_2$ and Cl$_2$_BrCl_Br$_2$ are focused over the oceans, there are larger changes seen on land in both cases than over the oceans—with the exception of the Arctic ocean, where ammonium aerosol drives a larger increase (Figure 2). The largest absolute increases in surface PM$_{2.5}$ in the OH–based experiments (not shown) are concentrated over northern hemisphere land including Europe, India, China, and Russia. Hydrogen peroxide experiments follow a similar pattern but have reduced impact over North America and greater increases in South America and southern Africa (Figure 2). The impact of daytime-only emissions (not shown) varied depending on the level of hydrogen peroxide emissions; for the H$_2$O$_2$_mid scenario, daytime-only emissions had higher PM$_{2.5}$ concentrations, but for the H$_2$O$_2$_low scenario the daytime-only emissions reduced PM$_{2.5}$ concentrations. Similarly to the tropospheric-wide changes, these changes are driven by increases in sulfate with decreases in nitrate partially compensating, particularly in the Cl$_2$ and Cl$_2$_BrCl_Br$_2$ methods. Surface organic aerosol also increases overall due to the increased atmospheric oxidation. Decreases in nitrate lead to small decreases in total PM$_{2.5}$ in some regions in the Cl$_2$ scenario. Overall, AOE exacerbates existing PM$_{2.5}$ air quality issues in populated regions of Europe and Asia (Figure 2).



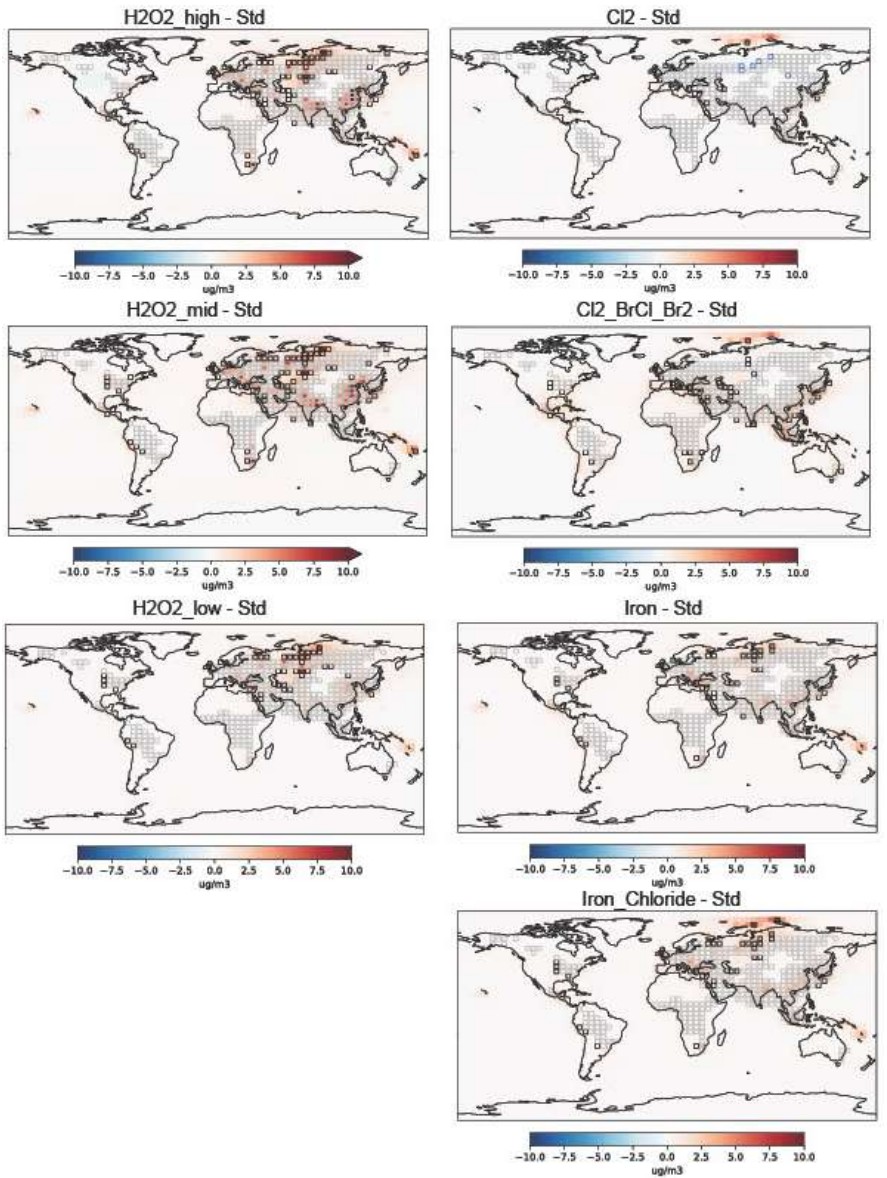


**Figure 2: Absolute change in surface PM$_{2.5}$ relative to the standard. Gray boxes: areas already in exceedance of USEPA's annual mean PM$_{2.5}$ primary standard of 9 μg m$^{-3}$, which remain in exceedance when the experiment is applied. Black boxes: the experiment brought a region from <9 μg m$^{-3}$ to an exceedance of ≥9 μg m$^{-3}$. Blue boxes: experiment led to improved PM$_{2.5}$ from an exceedance to < 9 μg m$^{-3}$.**






Figure 3 shows the absolute change in annual mean surface ozone across model experiments. In the global mean, surface ozone decreases on average in all methods (see Figure S4), with largest mean decreases in the $Cl_2\_BrCl\_Br_2$ and $H_2O_2\_high$ experiments. At the same time, individual grid boxes in the hydrogen peroxide experiments see increases in annual mean ozone across populated areas (see Figure 3). This has implications for the location of additional OH release,

balancing the targeting of methane point sources versus populated areas where ozone may increase. Figure 3 again highlights that $H_2O_2\_mid$ has worse air quality impacts for ozone than $H_2O_2\_high$.

Surface CO follows the tropospheric changes presented in Section 3.1.1 with decreases in hydrogen peroxide–based scenarios and increases in chlorine-based scenarios. Increases in surface CO concentrations in the chlorine-based scenarios are small relative to health guidelines and largely occur over the oceans. Surface CO decreases everywhere in the hydrogen

peroxide–based experiments.




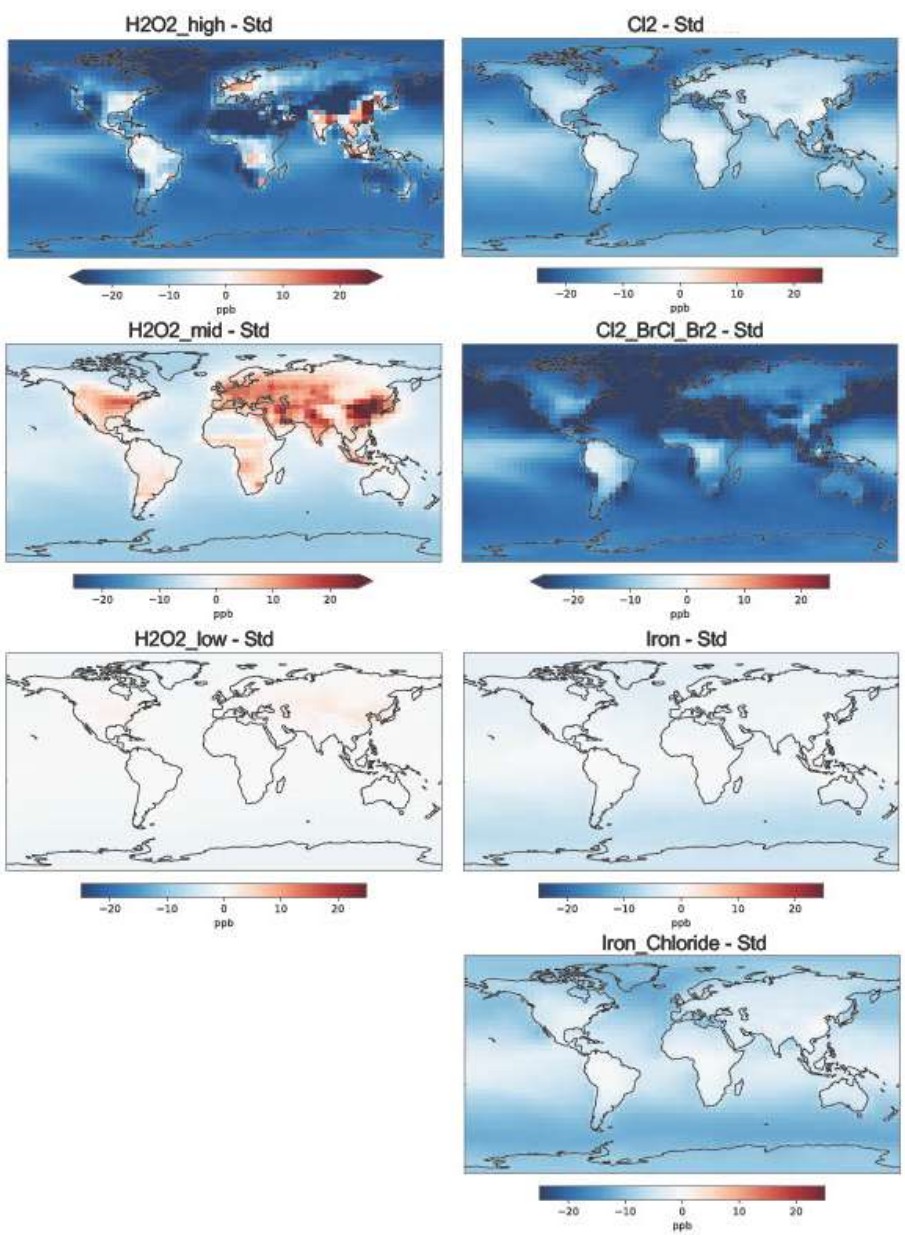

**Figure 3: Absolute change in surface ozone (ppb) between model experiments detailed in Table 1 and the standard model.**




### 3.1.5 Interactions with the Hydrogen Economy

Increases in hydrogen ($H_2$) emissions due to leakage from hydrogen applications will lead to a positive feedback on methane, as they react with OH and reduce the amount of OH available to oxidize methane (e.g., Bertagni et al., 2022; Ocko & Hamburg, 2022; Warwick et al., 2023). Based on results from a coupled Earth system model with interactive atmospheric chemistry, Warwick et al. (2023) find that increases in hydrogen alone lead to increases in methane, CO (whose main sink is also OH), and ozone. Hydrogen peroxide also increases (Warwick et al., 2023). A comparison with the results of the present study is shown in Table S8. A shift to hydrogen from fossil fuels, especially if the hydrogen is "green" (produced via renewable energy), will reduce emissions from fossil fuels including methane, CO, and $NO_x$. The net effect on methane thus depends on the amount of hydrogen produced and how it is produced (e.g., renewables or fossil fuels) as well as its leak rate (Bertagni et al., 2022; Ocko & Hamburg, 2022; Warwick et al., 2023).

How atmospheric oxidation enhancement may perform under a future global hydrogen economy also depends on how the hydrogen is produced and the leak rate. Based on the results of Warwick et al. (2023), a reasonable future global hydrogen economy scenario with a mid-range leak rate of 7% would lead to 1,000 ppb $H_2$ at the surface. Assuming 100% green hydrogen and thus concomitant reductions in fossil fuel emissions (including methane), OH concentrations decrease by 1% and the methane lifetime increases by 1.9% (Warwick et al., 2023). Without any reductions in fossil fuel emissions and a higher leak rate of 13% (1,500 ppb $H_2$ at the surface), OH concentrations decrease by 9.9% and the methane lifetime increases by 11.7% (Warwick et al., 2023). Reality will likely be somewhere in between. The effect of hydrogen emissions and emissions of hydrogen peroxide for atmospheric oxidation enhancement counteract each other (see Table S8). For chlorine, there is a synergistic effect on OH and CO.

At the same time, methane oxidation by OH eventually produces additional hydrogen (e.g., Bertagni et al., 2022). Based on Bertagni et al. (2022), the production of hydrogen from present methane oxidation from current OH could be approximately the same as future global hydrogen economy emissions with a 1% leak rate. In the current study, hydrogen is a species with globally fixed concentrations in GEOS-Chem, and thus this effect cannot be quantified. However, it is possible that the increased methane oxidation by OH in the hydrogen peroxide–based scenarios could lead to increased atmospheric hydrogen, which would then react away some of the additional OH. Thus, more hydrogen peroxide would need to be emitted for the same effect on methane found in the current study if both the two-way feedback of methane oxidation on hydrogen and vice versa and future hydrogen emissions from a global energy transition are considered. For the $Cl_2$ experiment in the current study that leads to a decrease in methane, OH is significantly reduced (by 28%), which would decrease the source of hydrogen from methane oxidation by OH. The effects are likely nonlinear, and thus it is not clear whether chlorine-mediated atmospheric oxidation enhancement would be less affected by the global hydrogen economy transition than OH-mediated atmospheric oxidation enhancement.



**4 Uncertainties**

      ***Model resolution.*** Here I use a coarse-resolution global model simulation. This is not appropriate to examine point source applications near high methane emitters. Model resolution can lead to biases due to nonlinear atmospheric chemistry.

However, these effects vary spatially and temporally and for tropospheric $NO_2$ are within ±8% of high-resolution simulations (C. Li et al., 2023). Resolution effects are most important in polluted regions and are becoming less important as anthropogenic sources are reduced (C. Li et al., 2023).

      ***Uncertainties in iron salt aerosol mechanism.*** Representation of chlorine release from iron salt aerosols in models remains highly parameterized. Complexities in iron solubility and speciation are not well represented. Depending on the

reaction rate and representation, a given mass of iron salt aerosol release may lead to increases or decreases in the methane lifetime. Additional laboratory studies of natural and engineered iron salt aerosol, including mixtures with ambient species, are needed to improve the understanding of the kinetics and driving factors. Field studies and additional observations that could help evaluate this mechanism in models (e.g., isotopic composition, chloride, chlorine, iron and its speciation) will also aid in constraining this process.

***Methods for OH release.*** The method and location of OH release will have different impacts. Here I only examined hydrogen peroxide as a mechanism for OH release, versus OH release without a specific mechanism. Effects of water vapor or artificial radiation for producing OH could have additional effects on climate forcers and air pollution.

      ***Chemistry and time horizon.*** Here I use the "simple" SOA mechanism (fixed-yield, direct, and irreversible formation); additional simulations would be needed to better understand the change in SOA production under varying

chemical regimes of OH and chlorine addition for methane oxidation. In addition, emissions are kept constant in year 2019. Most of the $PM_{2.5}$ effects predicted here are mediated by sulfate; future reductions in $SO_2$ emissions would likely limit this effect. Finally, uncertainties in the representation of halogen chemistry may impact the prediction of unintended consequences. For example, whether the halogen chemical mechanism includes interactions with sulfur seems to lead to different results for how increased chlorine affects sulfate aerosol between this study (increases) and Li, Meidan et al. (2023)

(decreases).

**5 Conclusions**

      Here we simulate multiple scenarios of enhancing atmospheric methane oxidation via OH or Cl including hydrogen peroxide release, $Cl_2$ emissions, and iron salt aerosol in a global chemical transport model to assess the potential for decreasing the methane lifetime and resulting impacts on other climate forcers, stratospheric ozone, and surface air quality.

The overall impacts of atmospheric oxidation enhancement methods on climate and human health involve multiple competing factors (see Figure 4).

      Based on the present work, current global demand of hydrogen peroxide is not sufficient to affect methane on a global scale, and this presents a challenge. Release of gas-phase $Cl_2$ is promising, but the exact mechanism to accomplish



this and ensure it is large enough to reduce rather than increase methane remains a major challenge. Two different model
approaches (this work, and Li, Meidan, et al., 2023) show that a large quantity of $Cl_2$ (>100–300 Tg/yr) must be added to the
atmosphere in order to have an impact. Smaller amounts, or increasing particulate chloride as shown in this study, will
increase methane. Here we find that emitting particulate iron alone to catalyze chlorine release from sea salt aerosol
following the reaction rate parameterization from Chen et al. (2024) does not release sufficient $Cl_2$ to decrease global
methane.

In addition to the impacts on methane, increased atmospheric oxidation via hydrogen peroxide- and chlorine-based
methods largely has a climate co-benefit, via increased tropospheric aerosols and decreased tropospheric ozone. Chlorine-
based methods increase other greenhouse gases and thus have reduced climate co-benefits.

Overall, ozone-depleting substances are increased in chlorine-based methods and decreased in OH-based methods.
Most of the species impacted are short-lived. Here I predict minimal reductions in stratospheric ozone, with the exception of
the chlorine experiment that includes bromine contamination. Longer simulations are needed to understand the full impacts
on stratospheric chemistry.

Chlorine-based methods reduce surface ozone air pollution. OH-based methods largely result in ozone reductions as
well but lead to increases in ozone in already polluted areas. Surface $PM_{2.5}$ pollution increases over land and in highly
populated regions across most chlorine scenarios and all OH-based methods, regardless of whether emissions are limited to
the oceans. These increases are on the order of present EPA air quality standards for annual mean $PM_{2.5}$.

Co-emission of bromine with chlorine appears to remove any benefit from chlorine-based approaches. In the case of
20% bromine release by mass with respect to chlorine, the methane lifetime nearly doubles. The addition of bromine also
generally results in worse outcomes with respect to surface air quality, halogenated greenhouse gases, and stratospheric
ozone. Additional experimental measurements of bromine species released from natural and engineered iron salt aerosol are
needed to constrain these effects.



**Figure 4: The overall impacts of atmospheric oxidation enhancement methods on climate and human health involve multiple competing factors. Note: * = sign of change depends on species (greenhouse gases [GHGs]) or location**
**(ozone).**

**Competing interests.** The contact author has declared that there are no competing interests.

**Acknowledgments**. I acknowledge helpful discussions with Jessica Haskins, Katherine Travis, and Qianjie Chen, and
funding from the National Academies of Science contract PO-10000845. Portions are reproduced with permission from the National Academy of Sciences, Courtesy of the National Academies Press, Washington, D.C.

**Data Availability**. The model data used in this study will be made available with a permanent DOI when the review process is complete through the Illinois Data Bank (https://databank.illinois.edu/) and will be subject to the terms of the Creative
Commons Attribution 4.0 International (CC BY 4.0). The standard model code is available at https://doi.org/10.5281/zenodo.5500717. For the review process, model data is temporarily available at the following link: https://www.dropbox.com/scl/fo/rtktddj7cru74p200x5q5/AI91DLtDyZJaXJlJEPY4j8Y?rlkey=x0yt2wjkxb9agv3gf4hwvh57 q&st=0l0vx6az&dl=0

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
