# Peer review of "Intended and Unintended Consequences of Atmospheric Methane Oxidation Enhancement"

_EGUsphere, 2024_

## Author Comment (AC2)

**Thank you to the reviewers for their thoughtful comments. Including your suggested revisions has improved the quality of the manuscript. My responses are indicated below in blue text. Line numbers refer to the track changes version of the manuscript.**

**The exact numbers and figures in the text have been updated to reflect an error in the original simulations in which natural dust emissions were inadvertently excluded, and errors in the calculation of methane lifetime against oxidation by tropospheric Cl.**

**These corrections have resulted in the comparison of impacts of the Chen et al. (2024) parameterization of $Cl_2$ release from natural iron salt aerosol on chlorine, ozone, and methane in GEOS-Chem to those in the van Herpen et al. (2023) much more similar (factor of 4 differences instead of factor of 100). The qualitative conclusions largely have not changed, with the exception that the iron salt experiments can lead to a decrease in methane of -2.5% to -8.3%, and these have been updated accordingly throughout the manuscript.**

**More detailed responses are below.**

By Matthew S. Johnson[1] and Maarten van Herpen[2]

1. Department of Chemistry, University of Copenhagen, Copenhagen, Denmark
2. Acacia Impact Innovation, Heesch, The Netherlands

**General Comments**

This paper uses a 3D atmospheric chemistry model to examine some of the methods that have been proposed for increasing methane oxidation in the atmosphere by enhancing concentrations of OH and Cl radicals, the two main radical sinks in today's atmosphere. The context for the paper includes the recent report by the U. S. National Academies of Sciences, A Research Agenda Toward Atmospheric Methane Removal [NASEM, 2024] which discusses Atmospheric Oxidation Enhancement, and the work was financed by a grant from the National Academies of Sciences, Engineering and Medicine. While the paper is significant and well written, and the model itself is sound, there are concerns with the assumptions and mechanisms used in the tested scenarios that lead to significant doubt regarding the conclusions. These include the use of the Chen parameterization for iron-salt aerosol, and the duration of the simulation. Besides these however, the assumptions are well described and largely  reasonable, appropriate scenarios have been used, the impact is significant, and overall the work is of high quality. There are issues, described below, that must be addressed before we can recommend publication. These include: 1. Discussing the impacts of the differences between the description achieved using a model with coarse resolution as in this work, and the chemistry arising from point source interventions such as a plume of $H_2O_2$ or $Cl_2$. 2. Refining and presenting key results and numbers, and including additional discussion of the error budget. 3. There is not consensus in the literature regarding some of the mechanisms and parameterizations employed, and their impact on the results should be described in more detail. 4. There are important additional papers in the field that should be referenced and included in the revised introduction and discussion. In general reviewers should try to refrain from citing their own work, with exceptions; as described below we believe this is

one of those cases. Overall this paper will be a welcome addition to the literature, pending revision to address the points raised below.

**Specific Comments**

The Abstract is well written. The opening sequence is strong, for example 'The chemistry involved is coupled; is nonlinear; and affects air quality, other greenhouse gases, and ozone-depleting substances.' In comparison the final sentence 'The overall impacts of atmospheric oxidation enhancement methods on climate and human health involve multiple competing factors.' is diffuse and anticlimactic. Please rewrite to clarify. If there is space, we suggest adding specific numerical results to the abstract.

**We clarify the sentence as follows: "The overall impacts of atmospheric oxidation enhancement methods on climate and human health involve not only their effectiveness at decreasing methane, but competing or complementary effects on other greenhouse gases and aerosol, as well as varying effects on surface air pollution." We add some numerical results regarding the iron salt aerosol experiments (see response to comment below).**

The abstract includes "I find that iron salt aerosol may not be effective at reducing methane on a global scale, depending on the reaction mechanism employed" and this conclusion should be deleted, because the parameterization used for ISA is likely underestimating $Cl_2$ production from iron-salt aerosol by 3 orders of magnitude (see explanation below). The way it is written here could confuse the reader making them think it is a trustworthy conclusion. Instead, we recommend to rephrasing this sentence to be specific about the mechanism, not about iron-salt aerosol.

**We have rephrased this sentence to reflect the corrected results as described earlier: "I find that larger emissions of iron salt aerosol are required relative to previous work to reduce methane on a global scale by at least a few percent ($\geq$565 Tg/yr), which indicates uncertainty in predicting the effectiveness of this method depending on the representation of the reaction mechanism and modeling framework employed."**

As well-described in the abstract 'The chemistry involved is coupled; is nonlinear..' Therefore more commentary is needed on the impacts or potential impacts of model resolution on the results. The model resolution is 4° latitude by 5° longitude, at the equator this is ca. 450 by 550 km. Many of the methods described in this paper are much more local. Examples include species that are lofted into the atmosphere from the surface by whirlwinds/convergences: Mineral dust is often stratified with variable dust densities. Similarly, sea spray aerosol, initially near the surface, is lofted high into the troposphere by local updrafts. Critically many of the interventions described here such as the addition of $Cl_2$ or $H_2O_2$ from a point source would occur as plumes with a high local concentration in the plume, that cannot be modeled accurately by taking a single average concentration over a grid cell hundreds of kilometers in each horizontal dimension, as, as was stated, the chemistry is coupled and nonlinear. Plumes will be lost to deposition before becoming mixed to the scale of this resolution. The effect is critical for a

chlorine intervention due to the interaction of Cl with NOx and the reaction of Cl with $O_3$, and the effects of NOx and ozone on OH production. Ozone has a short lifetime and is produced locally, in contrast methane has a very long lifetime and is mixed globally. This means that a chlorine intervention diluted to the model resolution will have to interact with many orders of magnitude more ozone than would take place in a real world intervention. The effect of this discrepancy between model resolution and plume dimension must be discussed, including an estimate of its impact on the results of the study. (A model of plume chemistry for high Cl conditions has been described in a preprint posted by Pennacchio et al [2024a].)

**We add reference to Pennacchio et al. (2025) [formerly 2024a] in the conclusions highlighting the uncertainty due to model resolution: "At the same time, Pennacchio et al. (2025) demonstrated the challenges in representing high-chlorine conditions in global lower-resolution models that result from point source emissions of iron for atmospheric methane oxidation enhancement, including how dilution of iron emission plumes and their interactions with the surrounding $NO_x$ and ozone gradients can change the direction of the change in methane predicted." (lines 636-639).**

**We clarify that the study referenced in lines 633-636 is specifically about resolution effects in GEOS-Chem.**

**We also add comparisons to two studies performed at higher resolution. "Another study using the GEOS-Chem model investigated daytime-only 600 m $H_2O_2$ towers in more detail over North America at a higher spatial resolution ($0.5° × 0.625°$ or ~50 km), and found that even widespread towers at emissions rates 10x higher than what is currently proposed would lead to negligible impact on global methane (Mayhew and Haskins, 2025)." (lines 138-142)**

**For chlorine, with respect to the natural iron salt mechanism: "Figures 2 and Figure S4 show the change in annual mean $Cl_2$ and Cl atom concentrations at the surface and zonally averaged through 20km, respectively, from adding the Chen et al. (2024) mechanism for $Cl_2$ production from aerosol iron photochemistry. The absolute increase in surface [Cl] (Figure 2) is largest over the North Atlantic ocean, with the largest relative increases over China reaching a factor of 3 (204%). In Chen et al. (2024), simulations were performed for December 9 - 31, 2017 over North China at 16 times higher spatial resolution ($0.25°$ latitude $× 0.3125°$ longitude) than this study ($4° × 5°$). They also added high-resolution anthropogenic fine-mode aerosol $Cl^-$ emissions in China from Fu et al. (2018) which are not included here and would further increase $Cl_2$ production through their mechanism (see equation 2). They found a maximum increase in [Cl] in an individual model gridbox of a factor of 20 to 40 which is consistent with the increased model horizontal resolution relative to my study." (lines 385-392)**

Another paper by Pennacchio et al [Pennacchio, 2024b] includes a section on Atmospheric Oxidation Enhancement in the context of feasibility and physical limitations to effective interventions. The Supporting Information includes a discussion of the yield of OH from $H_2O_2$ addition. Clearly the Horowitz approach is more sophisticated. How does the estimate of the OH yield in the two papers compare?

We include a comparison of the yield calculated in another GEOS-Chem study to that of Pennacchio et al. (2024) in lines 158-160: "Estimates include the previously described GEOS-Chem study which found approximately 30-60% of $H_2O_2$ on average is photolyzed, but this is highly spatially variable (Mayhew and Haskins, 2025), and a conservative theoretical estimate of 10% based on all potential chemical and physical pathways (Pennacchio et al., 2024)."

In this paper, the parameterization of the ISA mechanism describes chlorine production as a function of aerosol surface area density, $S$ (equation 2, section 2.2.2.3). This approach assumes that aerosol surface area is the relevant parameter. Another approach would be to base chlorine production on the concentration of the photoactive chromophore. Mikkelsen and coworkers [Mikkelsen, 2024] used experiment, quantum chemistry and an aqueous phase equilibrium model to investigate the ISA mechanism and demonstrate conclusively that the iron chloride chromophores are the key to production of chlorine. Although there could conceivably be a role for the surface area density in modulating chlorine release, Lim et al. [Lim, 2006] demonstrate chlorine production by shining light on a variety of bulk iron chloride solutions, showing that $Cl_2$ produced in bulk solutions escapes to the gas phase (with no dependence on the surface area of the solution). So it would seem that the Chen parameterization [20234] based on Wittmer [2015] depends on the conditions of their specific experiment and is difficult to generalize. In contrast an approach based on the concentration of the photoabsorbing species, such as iron(III) di- or tri-chloride, is more general.

Equation (2) is said to be based on Wittmer laboratory experiments, but when we compare Wittmer's experiments regarding the role of aerosol surface area (S) and [$Cl^-$], they do not agree with equation 2. Note that Wittmer varied the $Cl^-/Fe^{3+}$ ratio by changing [$Fe^{3+}$], so this ratio does not represent large changes in [$Cl^-$]. Wittmer [2015] included 3 'zero air' experiments where only [$Fe^{3+}$], $S$, and [$Cl^-$] were varied, and each of these experiments showed exactly the same $Cl_2$ production per $Fe^{3+}$ atom per unit time (referred to as lambda by Wittmer). Below we summarize these experiments:

| [$Cl^-$]/[$Fe^{3+}$] | $S$ ($10^{-2}$ m$^2$ m$^{-3}$) | [$Cl^-$] (mmol l$^{-1}$) |
|---|---|---|
| 101 | 2.5 | 29 |
| 51 | 3.2 | 30 |
| 13 | 3.0 | 37 |

From these Wittmer experiments, it is clear that if equation (2) were true, there should be a 50% difference in the Cl production per $Fe^{3+}$ atom between these experiments, but instead each experiment showed the same Cl production rate per Fe. Therefore, based on the Wittmer results, there is absolutely no reason to assume that [$Cl^-$] or $S$ should be included in equation 2.

The implication of using $S$ in the equation, if there should not really be a dependence on this parameter, is that it introduces a large effect when translating the laboratory experiment to real world conditions. Wittmer used an aerosol surface area of ca. $10^{-2}$ m$^2$/m$^3$, which corresponds to

ca. 30.000 $mm^2/cm^3$. In an ocean region, aerosol surface area density may be in the range of 10 – 60 $mm^2/cm^3$, which is at least 1000x lower. This could explain why the implementation of equation (2) reported in this article here has orders of magnitude lower Cl production than the one reported by van Herpen [2023].

**We add text highlighting uncertainties to the main text in lines 229-231: "The dependence on aerosol surface area concentration, and the impacts of pH and potential suppression of the rate by aerosol sulfate and organics (which are not considered in this mechanism) warrant future study."**

**We add a description of the model evaluation against hourly $Cl_2$ observations and the impact of the mechanism: "The Chen et al. (2024) mechanism was evaluated directly against hourly observations of gas-phase $Cl_2$ in Wangdu, China, as well as key parameters including aerosol [Cl⁻], aerosol total iron, $j_{NO2}$, and aerosol surface area (see equation 2). GEOS-Chem also captured the variability and magnitude of daily total Fe and Cl⁻ concentrations in fine-mode aerosol collected from 29 sites across North China, and slightly underestimated the observed solubility of aerosol Fe which was sampled at two sites, during the intensive study period at Wangdu (Chen et al., 2024). While adding the mechanism greatly improved the model performance of [$Cl_2$] at Wangdu, increasing simulated concentrations by a factor of 28 to 48, model concentrations remained underestimated. Chen et al. (2024) hypothesized this may be due to overestimated aerosol water in the model's thermodynamic module and thus underestimated aqueous-phase [Cl⁻] and [$Fe^{3+}$]." (lines 414-421)**

**We add text to the SI and Table S3 with more details on the calculations behind the Chen et al. (2024) mechanism: "Text S1: Additional details on calculations in the Chen et al. (2024) parameterization. Reaction rates given in the Wittmer et al. (2015b) Table 2 range from 6.6 – 8.7 × $10^{21}$ atoms $cm^{-2}hr^{-1}$ for artificial seawater and 8.7 – 13 × $10^{21}$ atoms $cm^{-2}$ $hr^{-1}$ for NaCl solution. Dividing by a factor of 3600 s/hr, the reaction rates are 1.8 – 2.4 × $10^{18}$ atoms $cm^{-2}s^{-1}$ for artificial seawater and 2.4 – 3.6 × $10^{18}$ atoms $cm^{-2}s^{-1}$ for NaCl solution. Wittmer et al. (2015b) summarized the rate as ~1.9 × $10^{18}$ atoms $cm^{-2}s^{-1}$ in the Abstract and Conclusions for a ratio of Cl⁻/$Fe^{3+}$ = 13. In Chen et al. (2024), from personal communication with Dr. Cornelius Zetzsch, this rate is incorrect as it was not adjusted for the volume of the chamber (requires dividing by the chamber size, 3.65 × $10^6$). After the correction, the rate is reduced to 5.2 × $10^{11}$ atoms $cm^{-2}s^{-1}$ (as used in Chen et al., 2024).**

**The value of α (see Table 1 and Section 2.2.2.3) can be calculated from the reaction rates and aerosol surface areas provided in Wittmer et al. (2015b (See Table S3 below). Across this range of Cl⁻/$Fe^{3+}$ and aerosol surface areas, the values calculated for α are within 14%. Chen et al. (2024) uses the value of α calculated from the summary rate and Cl⁻/$Fe^{3+}$ ratio reported in the Abstract of Wittmer et al. (2015) rounded to two significant figures (α =1.4 × $10^5$ $\mu m^{-2}$ $M^{-2}$).**

**Table S3 Summary of Chen et al. (2024) parameter calculations from values in Wittmer et al. (2015b) and personal communication with Dr. Cornelius Zetzsch.**

| $Cl^-/Fe^{3+}$ | Reported rate (atoms $cm^{-2}s^{-1}$) | Corrected rate (atoms $cm^{-2}s^{-1}$) | Aerosol surface area concentration ($\mu m^2/cm^3$) | Calculated $\alpha$ ($\mu m^{-2}$ $M^{-2}$) |
|---|---|---|---|---|
| 13 | $1.9 \times 10^{18}$ | $5.2 \times 10^{11}$ | 30000 | $1.36 \times 10^5$ |
| 101 | $2.8 \times 10^{17}$ | $7.7 \times 10^{10}$ | 18000 | $1.55 \times 10^5$ |

"

The article under review also identifies this discrepancy on page 13, lines 322-330, where it compares the impact of the two models on methane oxidation (0.2% with the Chen parameters versus 20% in the van Herpen parameterization, for methane oxidation by mineral dust). This factor of 100 difference can be explained as being due to the impact of using the $S$ parameter (dust will increase $S$, but not to the levels used by Wittmer). Note that the article calculated 'up to 10 times higher' in line 331, but this should be 'at least 100x higher'.

**These results were impacted by the inadvertent lack of natural dust emissions in the original simulations and an error in the calculation of methane lifetime against oxidation by tropospheric Cl. These numbers have been updated: increase in methane loss of 5.4% (factor of 3.7 smaller) and decrease in tropospheric ozone burden by -0.18% (factor of 4 smaller).**

Another way to compare van Herpen with the Chen parameterization is to consider that iron-salt aerosols produce $Cl_2$ in the model. This implies that the $Cl_2$ and the Iron scenario are in principle the same, but different amounts of $Cl_2$ are emitted. The $Cl_2$ scenario emitted 1250 Tg $Cl_2$ to increase the Cl burden by 2185%, while the iron scenario emitted 565 Tg Fe to increase the Cl burden by 179%. Assuming a linear relation between $Cl_2$ and Cl (considering the majority of $Cl_y$ is $Cl_2$), this suggests that the iron scenario emitted 8.2% as much $Cl_2$ compared to the $Cl_2$ scenario, thus 102.5 Tg $Cl_2$. This corresponds to 0.18 g $Cl_2$ per g Fe emission, while van Herpen found 70 g $Cl_2$ per g Fe emission *per day* (and multiplied by an average 5 days lifetime is 350 g $Cl_2$ per g Fe). This leads to a difference between van Herpen and Chen of up to 3 orders of magnitude.

**We have added figures showing the change in $Cl_2$ and Cl concentrations from adding the Chen et al. (2024) mechanism alone (now Figures 2 and S4) and text describing them in lines 385-388: "Figures 2 and S4 show the change in annual mean $Cl_2$ and Cl atom concentrations at the surface and zonally averaged through 20km, respectively, from adding the Chen et al. (2024) mechanism for $Cl_2$ production from aerosol iron photochemistry. The absolute increase in surface [Cl] (Figure 2) is largest over the North Atlantic ocean, with the largest relative increases over China reaching a factor of 3 (204%)."**

**We also have added a Table S7 to the SI with the tropospheric-wide changes in $Cl_2$ burdens compared to the Cl burdens, their relative changes from the standard model in each experiment, and the ratio of the changes, as well as text discussing this:**

**Table S7 Comparison of changes in tropospheric-wide $Cl_2$ and Cl burdens. Note: Percent change in annual mean tropospheric burdens for model experiments given in parentheses are relative to the standard version (\* = relative to standard + Chen et al., 2024).**

| | $Cl_2$ (Gg) | $\Delta Cl_2$ (%) | Cl (kg) | $\Delta Cl$ (%) | $\Delta Cl_2/\Delta Cl$ ratio |
|---|---|---|---|---|---|
| **Standard** | 1.06 | -- | 313 | -- | -- |
| **Cl2** | 926.7 | 87369.7% | 7250 | 2213.6% | 39.5 |
| **Cl2_BrCl_Br2** | 829.7 | 78216.5% | 6171 | 1869.5% | 41.8 |
| **Standard+Chen** | 1.16 | 9.8% | 343 | 9.4% | 1.0 |
| **Iron\*** | 14.04 | 1107.6% | 924 | 169.4% | 6.5 |
| **Chloride\*** | 3.99 | 243.2% | 955 | 178.6% | 1.4 |
| **Iron_Chloride\*** | 34.30 | 2849.4% | 2599 | 658.0% | 4.3 |

**We have added a figure showing the zonal mean change in $Cl_2$ and Cl concentrations from adding iron emissions in the Iron scenario (Figure 3). We added text to highlight that there is not a 1:1 relationship between the changes in Cl and $Cl_2$, and that HCl is the largest component of $Cl_y$ in all model experiments:**

**"Figures 3 and S4 show that the additional release of $Cl_2$ due to natural or emitted iron leads to changes in $Cl_2$ and Cl that have differing spatial patterns and magnitudes in the annual mean due to physical and chemical processing. Tropospheric-wide differences between changes in $Cl_2$ and changes in Cl (see Table S7) are minor from adding the Chen et al. (2024) parameterization alone or in the Chloride experiment, but diverge most in experiments where iron emissions are added (Iron and Iron_Chloride). There, the relative increase in $Cl_2$ is 4 to 6.5 times greater than what is realized for Cl (Table S7)." (lines 430-435)**

**"Within each experiment, the longer-lived reservoir species hydrogen chloride (HCl) contributes the largest mass to the total $Cl_y$ burden." (lines 285-286)**

**"As in the hydrogen peroxide experiments, HCl still remains the dominant component of $Cl_y$." (lines 301-302)**

We also note that the van Herpen parameterization has been validated against observations in the real atmosphere and against laboratory observations, and it agrees with the cycling rates reported by Wittmer. While there is no discussion about the validity of the Chen parameterization in the paper it would clearly not agree with observations if it results in 3 orders of magnitude lower $Cl_2$ production.

**We add discussion of the evaluation of the Chen et al. (2024) parameterization against observations (lines 414-421): "The Chen et al. (2024) mechanism was evaluated directly against hourly observations of gas-phase $Cl_2$ in Wangdu, China, as well as key parameters including aerosol [Cl-], aerosol total iron, $j_{NO2}$, and aerosol surface area (see equation 2). GEOS-Chem also captured the variability and magnitude of daily total Fe and Cl- concentrations in fine-mode aerosol collected from 29 sites across North China, and slightly underestimated the observed solubility of aerosol Fe which was sampled at two sites, during the intensive study period at Wangdu (Chen et al., 2024). While adding the mechanism**

**greatly improved the model performance of [$Cl_2$] at Wangdu, increasing simulated concentrations by a factor of 28 to 48, model concentrations remained underestimated. Chen et al. (2024) hypothesized this may be due to overestimated aerosol water in the model's thermodynamic module and thus underestimated aqueous-phase [$Cl^-$] and [$Fe^{3+}$]."**

**Due to the aforementioned corrections, the difference is a factor of 4 tropospheric-wide (this has been corrected in the manuscript).**

Regarding equation (2), the author should also better clarify how the parameters [$Cl^-$] and [$Fe^{3+}$] are defined. Are these aqueous phase concentrations (so for example the concentration of $Fe^{3+}$ in the aerosol), or are they a concentration per volume of air? Please provide some typical numbers for these concentrations, so that it is possible to make a calculation and check the validity of this equation.

**We clarify the units to be more clearly aqueous-phase concentrations, changing "mol l$^{-1}$" to "mol l$^{-1}$ water, or M" (see lines 205-206).**

**We add text referencing the model evaluation of aerosol total iron, iron solubility, and aerosol Cl- to the text (see response to comment above this one).**

Finally, the paper is only using accumulation mode $Fe^{3+}$, while Fe photochemistry has been shown to also occur in large particles. What is the rationale for only including accumulation mode $Fe^{3+}$? If the aerosol mix and grow in size, the model would exclude them from the calculation of chlorine production. Clearly this is not the case in reality. How much chlorine production is being excluded by the model?

**We add text explaining the rationale (lines 225-229): "For consistency with the version of the Chen et al. (2024) mechanism that was evaluated against observations (see Section 3.1.2), I also include this process on fine-mode aerosols only; the chamber experiment on which the mechanism is based used fine mode aerosol (Wittmer et al., 2015b) and the lifetime of smaller particles is longer relative to coarse particles such that iron and chloride have time to mix within the aerosols before being deposited (Moffet et al., 2012, Zhu et al., 2022)."**

In section 2.2.1.1, a test is made of the impact of releasing the total annual production of $H_2O_2$ into the atmosphere, 4.1 Tg($H_2O_2$)/year. Although the impact (we assume on global methane concentration, please clarify) was negligible, it would be interesting to learn the yield from the model. What is the mole fraction of change in methane to change in $H_2O_2$? What fraction of the $H_2O_2$ resulted in additional OH in the atmosphere, and what fraction of that OH reacted with $CH_4$? To put the intervention into perspective, what is the current annual natural production of $H_2O_2$ in the atmosphere? Also, the paper states that releasing $H_2O_2$ at altitude, or only during the day, resulted in increased methane oxidation. Please quantify this increase, for example through the yield of methane removed per $H_2O_2$ emitted. Also, if $H_2O_2$ release at altitude and during the day led to increased methane removal, please explain why this approach was not chosen for the $H_2O_2$ scenarios.

**We clarify the text as follows: "All 4.1 Tg/yr scenarios produced negligible effects on the methane lifetime." (line 133).**

**We add reference another GEOS-Chem study that implemented the complex tracking and calculations needed to answer the yield- and branching-related questions: "Another study using the GEOS-Chem model investigated daytime-only 600 m $H_2O_2$ towers in more detail over North America at a higher spatial resolution (0.5° × 0.625° or ~50 km), and found that even widespread towers at emissions rates 10x higher than what is currently proposed would lead to negligible impact on global methane (Mayhew and Haskins, 2025). This is in part because the fraction of $H_2O_2$ converted to OH is ~30-60% but is driven by the small fraction of the produced OH that reacts with methane (23% at the most and frequently much less) (Mayhew and Haskins, 2025)." (line 137-142)**

In section 2.2.1.2, please state the $H_2O_2$ photolysis lifetime (2 days?), this may help readers understand why limiting $H_2O_2$ emissions to daytime had negligible effects and why the OH yield from this source is as low as it is. A qualitative statement is made, 'Not all hydrogen peroxide is immediately photolyzed to produce OH and may undergo alternate reactions.', which could be better understood by providing this value.

**Here I add text following this sentence: "Estimates include the previously described GEOS-Chem study which found approximately 30-60% of $H_2O_2$ on average is photolyzed, but this is highly spatially variable (Mayhew and Haskins, 2025), and a conservative theoretical estimate of 10% based on all potential chemical and physical pathways (Pennacchio et al., 2024)." (lines 158-160)**

In section 2.2.2.1 on emission of chlorine, it would be useful to compare the emission scenarios to the present annual production of $Cl_2$, as was done for $H_2O_2$. We found one reference which indicates production is about 60 Tg($Cl_2$)/year (https://www.chlorineinstitute.org/chlorine-manufacture).

**Thank you for providing this helpful information! I found another reference with a similar amount (58 Tg/yr) and added this information: "This is approximately 20 times higher than the current total tropospheric source of gas-phase inorganic chlorine in GEOS-Chem (54 Tg/yr; Wang et al., 2021) or the current manufacture of $Cl_2$ (58 Tg/yr; World Chlorine Council)." (lines 177)**

We are wondering about the formation of air pollution as a consequence of the chlorine intervention. A simplistic 'zero sum game' point of view would be that there is a certain yield of $O_3$ and PM (smog) from the amount of VOC that is emitted, and that atmospheric oxidation enhancement (AOE) would merely change the location of the smog formation, without changing the amount of smog formation. This is rather like the analysis that cloud seeding would potentially only change the location of the rainfall without changing the total rainfall. If this is the case then the formation of air pollution from the introduction of Cl or OH could be seen as 'a feature not a bug', as smog formation could be triggered to occur over unpopulated areas and away from sensitive ecosystems (for example smog impacts on land plants is larger than the

impact of smog occurring over the oceans).

**We add text highlighting prior work on how aerosol formation can be oxidant-limited: "These results are consistent with prior work suggesting that aerosol production can be oxidant-limited (e.g., Mayhew and Haskins, 2025; Shah et al., 2018)." (lines 551-552)**

The picture becomes more complicated when methane is added to the mix. In contrast to the VOCs, it's long lifetime means it is well mixed globally. A reduction in methane means air quality will improve globally as less ozone will be made. It is difficult to see how this interaction can be described in a model that only runs for one year. This issue would seem to interfere with the ability of the study to make conclusions concerning certain aspects of air quality such as inorganic aerosols and SOA. While the model predicts an instantaneous change, one would also like to know the steady state change, as could be derived in the multiple year models such as Li [2023] and Meidan [2024]. This is not possible when methane has been fixed. Therefore, we recommend removing the sections that discuss air pollution impact on PM2.5 and inorganic aerosols. If the author decides to keep it in, please discuss the issue in the comparison with the Meidan and Li results. Moreover, the model in its current form will likely overpredict air pollution because VOCs and DMS do not have time to decrease, they are always at or near peak, while at the same time methane is fixed.

**Throughout reference to the surface air quality changes in the text and figures, we add text highlighting that this is for year 2019 only. We further add text in lines 557-564: "These results represent one single year of AOE application (2019). As such, the exact magnitude and spatial patterns of changes in annual mean $PM_{2.5}$ due to AOE may vary interannually due to variability in meteorology and its effects on natural emissions, pollutant transport, and $PM_{2.5}$ removal, and the variability in background anthropogenic emissions. Moreover, long-term simulations using the CESM2 model suggest that relative changes in tropospheric sulfate aerosol and ozone due to $Cl_2$ gas-based AOE increase during the first 15 years of continuous application and then stabilize (Li, Meidan et al., 2023). Here, Figure 2 highlights the potential risks in surface $PM_{2.5}$ air quality in different regions in the initial years of deployment."**

**We rephrase to clarify the effects on ozone: "In the mean across all scenarios, there are declines in surface ozone air pollution (Figure S6)." (line 546)**

The same argument applies to the discussion of changes in CO (page 19), which would also be temporarily increased until longer-lived precursors ($CH_4$ and VOCs) are allowed to stabilize.

**We add text: "These changes represent the short-term impact over 1 year of applied AOE." (line 592)**

The scenario described in Section 2.2.2.2 on bromine contamination describes the impact of emitting bromine at a mass fraction corresponding to 20% of chlorine emission (or 9% as mole fraction). Reference is given to experiments by Wittmer et al. on Br and Cl production from artificial seawater, which found that the mass fraction of Br to Cl ranged from 0 to 2.5, and to

experiments where undetectable traces of Br in a salt pan resulted in 1:1 emission of Br. Based on this evidence, the factor of 20% is chosen for the simulation. Note however that the experiments measured initial emission of bromine, during a phase of the experiment where Br has not yet been depleted from the aerosols. Bromide is a minor component of seawater occurring at a Br/Cl mass fraction of 0.35% or mole fraction of 0.154%. Studies show that bromide ions are substantially depleted in sea-salt aerosols in the field [Sander, 2003; Saiz-Lopez, 2006], which implies that after the original Br is depleted, it's chemistry is no longer relevant -- there is simply not much Br there relative to Cl. Thus, in real world situations, bromine will be depleted very rapidly and the net result will not be 20% but something about 100 times lower, with a steady state more closely corresponding to the ratios found in seawater. This is especially true when aerosol chloride becomes depleted to the point that re-uptake of HCl from the gas phase to the particle phase becomes the most important source of chloride for ISA. How do the results change if a realistic Br$^-$ fraction is used? Please modify text and comment accordingly.

**We add text to clarify: "Here I assume that of the total desired chlorine release (1,250 Tg/yr as in the Cl$_2$-ocean simulation), 20% of that by mass of bromine is released in equal parts Br$_2$ and BrCl (resulting in 1193 Tg/yr Cl$_2$, 187 Tg/yr Br$_2$, and 129 Tg/yr BrCl). This scenario represents a bounding case if artificial sea salt containing bromine impurities were to be continuously emitted as part of an AOE method. In prior work, increasing the flux of natural sea salt aerosol in GEOS-Chem led to relative increases in tropospheric-wide reactive bromine that were comparable to that of reactive chlorine (Horowitz et al., 2020)." (lines 195-200).**

Section 2.2.2.3 describes the Iron Salt Aerosol scenario, including the alpha factor which scales Cl$_2$production using the surface area concentration. As noted previously there does not seem to be a convincing physical mechanism behind this parameterization as chlorine production is due to absorption of light by the iron chloride chromophore, and re-oxidation of Fe(II) by H$_2$O$_2$, and doesn't involve the surface area. The chromophore concentration depends on the volume of liquid and the concentration of the iron chlorides. An alternative mechanism is described in the paper by van Herpen et al [2023] and validated using the results of field experiments [Zhu, 1993; Mak, 2003; van Herpen, 2023]. It is clear that the two methods are not in agreement. Although Table S4 provides a descriptive overview of the two models, the paper does not include an analysis of why the Chen model was chosen. Critically, the Chen model seems unphysical and requires much more iron, resulting in an inaccurate assessment of the environmental impact of ISA. The paper concludes (Line 332) that 'With the parametrization used in this study, iron salt aerosol cannot produce enough chlorine to overcome the decrease in methane loss via the OH channel.' This conclusion is dependent on the seemingly unphysical and erroneous choice of parameterization of the ISA mechanism.

**We add text discussing the evaluation of Chen et al. (2024) against a suite of field observations as previously described. We also add text to the SI further explaining the Chen et al. (2024) parameterization and the calculations of α which are consistent across a range of surface areas and Cl/Fe ratios:**

"Text S1: Additional details on calculations in the Chen et al. (2024) parameterization. Reaction rates given in the Wittmer et al. (2015b) Table 2 range from $6.6 - 8.7 \times 10^{21}$ atoms $cm^{-2}hr^{-1}$ for artificial seawater and $8.7 - 13 \times 10^{21}$ atoms $cm^{-2}$ $hr^{-1}$ for NaCl solution. Dividing by a factor of 3600 s/hr, the reaction rates are $1.8 - 2.4 \times 10^{18}$ atoms $cm^{-2}s^{-1}$ for artificial seawater and $2.4 - 3.6 \times 10^{18}$ atoms $cm^{-2}s^{-1}$ for NaCl solution. Wittmer et al. (2015b) summarized the rate as $\sim 1.9 \times 10^{18}$ atoms $cm^{-2}s^{-1}$ in the Abstract and Conclusions for a ratio of $Cl^-/Fe^{3+} = 13$. In Chen et al. (2024), from personal communication with Dr. Cornelius Zetzsch, this rate is incorrect as it was not adjusted for the volume of the chamber (requires dividing by the chamber size, $3.65 \times 10^6$). After the correction, the rate is reduced to $5.2 \times 10^{11}$ atoms $cm^{-2}s^{-1}$ (as used in Chen et al., 2024).

The value of $\alpha$ (see Table 1 and Section 2.2.2.3) can be calculated from the reaction rates and aerosol surface areas provided in Wittmer et al. (2015b (See Table S3 below). Across this range of $Cl^-/Fe^{3+}$ and aerosol surface areas, the values calculated for $\alpha$ are within 14%. Chen et al. (2024) uses the value of $\alpha$ calculated from the summary rate and $Cl^-/Fe^{3+}$ ratio reported in the Abstract of Wittmer et al. (2015) rounded to two significant figures ($\alpha = 1.4 \times 10^5$ $\mu m^{-2}$ $M^{-2}$).

Table S3 Summary of Chen et al. (2024) parameter calculations from values in Wittmer et al. (2015b) and personal communication with Dr. Cornelius Zetzsch.

| $Cl^-/Fe^{3+}$ | Reported rate (atoms $cm^{-2}s^{-1}$) | Corrected rate (atoms $cm^{-2}s^{-1}$) | Aerosol surface area concentration ($\mu m^2/cm^3$) | Calculated $\alpha$ ($\mu m^{-2}$ $M^{-2}$) |
|---|---|---|---|---|
| 13 | $1.9 \times 10^{18}$ | $5.2 \times 10^{11}$ | 30000 | $1.36 \times 10^5$ |
| 101 | $2.8 \times 10^{17}$ | $7.7 \times 10^{10}$ | 18000 | $1.55 \times 10^5$ |

"

As previously discussed, the results have been corrected due to two errors, which affected the impact of iron salt aerosol. The sentence referenced above has been corrected and moved, and the discussion altered as follows (lines 436-454):

"With the parametrization used in this study, iron salt aerosol can lead to changes in steady-state methane of -8.3% to +2.5% depending on the emissions employed and their relative effects on Cl vs. OH (see Figure 1). The Iron experiment led to a small decrease in steady-state methane (–2.5%) from a tropospheric-wide factor of 2.7 increase in Cl atom (see Table S7). The Iron_Chloride experiment led to a larger decrease in steady-state methane (-8.3%) due to a larger 7.6-fold increase in Cl atom burden, despite a much larger reduction in OH (-8.6% vs. -2.1%; see Table 2 and Table S7). For a similar increase in Cl atom burden, emitting chloride (Chloride, 2.8-fold increase in Cl) instead of iron aerosol (Iron, 2.7-fold increase in Cl) led to a larger decrease in OH (-3.9% vs. -2.1%; see Table 2) and hence net increase in methane (+2.5%). Li, Meidan et al. (2023) found that a 2.8-fold increase in Cl burden (from 88 Tg/yr gas-phase $Cl_2$ emission) was insufficient to decrease methane, while a 7.9-fold increase in Cl burden (from 313 Tg/yr gas-phase $Cl_2$ emissions) overcame the OH competition and led to a decrease in methane concentrations by about 6% after 10 years (Li, Meidan, et al., 2023). This suggests that the threshold of additional chlorine needed to

**overcome the OH limitation depends on what is emitted and the background chemistry in the model employed.**

**Here I find that emitting sea salt chloride along with particulate iron increases methane loss. This is a function of the formulation of the Chen et al. (2024) mechanism, which occurs on sea salt chloride aerosol and increases with increasing [Cl⁻] concentrations (see Section 2.2.2.3 and Equation 2). Artificial chloride aerosol emissions in addition to the particulate iron emissions can replenish the sea salt chloride that was converted to $Cl_2$ and increase aerosol [Cl⁻] concentrations, leading to greater overall production of $Cl_2$ (see Table S7).**

How would correction for the model resolution problem, by inclusion of a proper description of the much more concentrated chemistry occurring in a plume, affect the conclusion 'With the parametrization used in this study, iron salt aerosol cannot produce enough chlorine to overcome the decrease in methane loss via the OH channel.'? Please consider and modify as may be needed. It seems that such a general categorical conclusion is not supported given the assumptions and errors.

**See response to the comment above and earlier response regarding additional discussion of model resolution uncertainties and in the Uncertainties section.**

Line 314, it is not clear how the Cl + $CH_4$ reaction is air-density dependent? The reaction rate coefficient $k$ is temperature dependent, that is true, but it is not air-density dependent. The rate of methane change will be given by $r = -d[CH4]/dt = k[Cl][CH4]$. Here the rate $r$ depends on the concentrations of the species, but $k$ does not. Please rewrite to clarify.

**We clarify as follows: "Differences in our results may be due to differences in the vertical distribution of Cl as well as the meteorology between the two models, as the Cl + $CH_4$ reaction is temperature-dependent (see Table S6) and the impacts of local changes in the reaction rate on the global methane lifetime are weighted by air density." (lines 374-377).**

Line 342, check 'Here I find that emitting sea salt chloride instead of or along with particulate iron worsens methane outcomes.' and modify depending on circumstances. Also note that sea spray aerosol is ubiquitous in the marine environment so under those circumstances chloride is abundant.

**We modify this discussion based on the corrected results (see also response to previous comment): "Here I find that emitting sea salt chloride along with particulate iron increases methane loss. This is a function of the formulation of the Chen et al. (2024) mechanism, which occurs on sea salt chloride aerosol and increases with increasing [Cl⁻] concentrations (see Section 2.2.2.3 and Equation 2). Artificial chloride aerosol emissions in addition to the particulate iron emissions can replenish the sea salt chloride that was converted to $Cl_2$ and increase aerosol [Cl⁻] concentrations, leading to greater overall production of $Cl_2$ (see Table S7)." (Lines 448-453)**

It looks like atmospheric hydrogen ($H_2$) is only discussed on page 21, in section 3.1.5. This interlude is disconnected from the rest of the paper, superficial and not connected to the

modelling work (the manuscript states that in the model hydrogen is fixed and so it is not possible to examine hydrogen-related questions) and without clear conclusions, and we recommend that it be cut from the manuscript. Perhaps it could be developed into a future publication. The section begins by claiming that increases in hydrogen emissions will lead to a positive feedback on methane. If we see widespread adopttion of hydrogen as an energy carrier in the future,presumably it will replace carbon based energy carriers, leading to a decrease in methane emissions from natural gas and fossil related sources. This claim seems doubtful and unsupported by evidence. It is unclear if 'positive feedback on methane' and 'increases in methane' refer to methane lifetime or atmospheric concentration. Also it is not immediately clear whether addition of $H_2$ would lead to increased OH or decreased, as much of the $H_2$ would presumably be emitted in high-NOx northern hemisphere conditions where more OH is produced by $HO_2$ + NO than from ozone, and also, given the lifetime of $H_2$, most of it will be oxidised in the hemisphere in which it is emitted.

**We remove this section and Table S8 in the supplement. We add a brief reference to Section 4 (Uncertainties – Chemistry and time horizon) regarding those resulting from interactions with the hydrogen chemistry and the hydrogen economy:**

**"We do not include interactive hydrogen ($H_2$) chemistry in our model simulation. Increased methane oxidation by OH in the hydrogen peroxide–based scenarios could lead to increased atmospheric $H_2$, as would potential future increases in hydrogen applications and their associated $H_2$ emissions from leakage. This would lead to a positive feedback on methane, as $H_2$ reacts with OH and reduces the amount of OH available to oxidize methane (e.g., Bertagni et al., 2022; Ocko & Hamburg, 2022; Warwick et al., 2023)." (lines 653-657)**

The discussion of uncertainties needs to include much more detail in order to increase usefulness to other researchers - whether their goal is to use the results (and then they need to understand the uncertainties), or to find ways to reduce the uncertainties using models, field studies or laboratory work. The text states that the use of a coarse resolution global model is not appropriate for point source applications near high methane emitters. Neither is it appropriate for point source applications not near high methane emitters. IWe recommend that this section also states what errors will arise from the use of a coarse grid, and how large these errors could potentially be. The discussion of the uncertainties in the ISA mechanism glosses over the issues, that a model was chosen that is based on a questionable mechanism and parameterization, and ignores available field studies. Overall, it is better to give details or leave out the discussion if it is only superficial.

**We add discussion to the Uncertainties section regarding the resolution. "At the same time, Pennacchio et al. (2025) demonstrated the challenges in representing high-chlorine conditions in global lower-resolution models that result from point source emissions of iron for atmospheric methane oxidation enhancement, including how dilution of iron emission plumes and their interactions with the surrounding $NO_x$ and ozone gradients can change the direction of the change in methane predicted."  (lines 635-639)**

In the conclusions, please be sure to come back to the expectations that the title, 'Intended and Unintended Consequences of Atmospheric Methane Oxidation Enhancement' engendered in the

reader. Make a clear summary of the consequences - the present conclusion lacks the hard edge it ought to have.

**We add a sentence to the end of the conclusions: "Overall, additional research in higher-resolution, longer-term modeling frameworks as well as laboratory experiments is needed to constrain whether the AOE methods have the desired intended consequences of sufficiently decreasing atmospheric methane, and the risk level for the unintended consequences of potential increases in halogenated greenhouse gases, ozone-depleting substances, and particulate matter air pollution."**

Overall, there are a lot of things to like in this article and it is the nature of reviewing to mainly comment on the aspects that the reviewers believe could or should be improved.

**Technical Corrections**

Line 13, '..depending on the reaction mechanism employed.' Check word choice, the mechanism is ISA, but the result will depend on the parameterization used in the model. The sentence suggests that a comparison was made between different parameterizations. This would be good to do, but then the author should also run the model with the van Herpen parameterization.

**This sentence is updated as previously described: "I find that larger emissions of iron salt aerosol are required relative to previous work to reduce methane on a global scale by at least a few percent ($\geq$565 Tg/yr), which indicates uncertainty in predicting the effectiveness of this method depending on the representation of the reaction mechanism and modeling framework employed."**

Note the important distinction between chloride (Cl-) and chlorine (Cl(0), e.g. Cl and $Cl_2$). Check usage, be consistent and specific.

**Checked and added clarifications as discussed in previous response.**

Line 24. Please add a reference to the National Academies of Sciences, Engineering and Medicine (NASEM) report [2024] 'A Research Agenda Toward Atmospheric Methane Removal'.

**Added.**

Line 25, the manuscript states that tropospheric chlorine is responsible for 1–5% of methane removal, with reference to five papers on methane removal. van Herpen et al. [2023] give this range as 0.8 to 3.3%, with reference to five papers on atmospheric chlorine. There seems to be a discrepancy and it is important, as the Cl reaction has a large kinetic isotope effect affecting models of methane emissions sources [Röckmann 2024]. Please double check the numbers and use primary sources when possible.

**Thank you for pointing this out! The existing references were in the wrong part of the sentence, which we move, and we add references which include both modeling and observational studies that were inadvertently missing. We update the lower bound to 0.23%**

**following Gromov et al. (2018) and its interpretation in the latest methane budget report (Saunois et al., 2025). : "These include processes to decrease the atmospheric lifetime of methane by enhancing its main sinks (e.g., Abernethy et al., 2023; Gorham et al., 2023; Li, Meidan, et al., 2023; Ming et al., 2022; Wang et al., 2022), using oxidation by tropospheric OH (currently >90%) and tropospheric Cl (currently 1–5%)(Allan et al., 2007; Gromov et al., 2018; Hossaini et al., 2016; Platt et al., 2004; Wang et al., 2019, 2021)." (lines 27-30)**

Line 32, add reference to NASEM report [2024]-- one of the key conclusions is 'For example, a technology gap exists in which no commercial mitigation technologies oxidize methane at concentrations below 1,000 parts per million (ppm) even though most methane emissions are found at concentrations closer to 2 ppm.' Add reference to Pennacchio [2024b] which concludes that there are considerable physical and practical constraints to currently available technologies.

**We add these references.**

Line 33, no reference is given for OH generators on a smaller scale. Such systems are described in e.g. Meusinger [2017] and Johnson [2014].

**We add a reference here.**

Line 39, Technically, these experiments demonstrated release of molecular chlorine $Cl_2$ not chlorine atoms. In addition to these references, note that the wavelength dependence of the release of $Cl_2$ from sodium/iron salt samples is described by Mikkelsen et al. [2024]

**Wittmer et al. (2015a,b) both quantify atomic chlorine production using the radical clock method. We add reference to Mikkelsen et al. (2024).**

Line 72, note that the chemical form of the iron will have a large impact on its activity. Iron could be metallic particles, mixed iron oxides/hydroxides, iron chlorides, iron complexes with organic molecules, iron locked in minerals, and so on.

**We add a sentence: "The speciation and solubility of iron in GEOS-Chem is discussed in Section 2.2.2.3." (now lines 78-79)**

Line 97, It is impressive that the 2019 annual mean surface methane concentration is known to so many digits of accuracy, 1866.58 ppb. It may be useful to note the range of values that are encountered in the atmosphere through the course of one year, ca 20 ppb.

Line 168, change 'that even when Br atom was below the detection..' to 'that even when Br atoms were below the detection..'

**Changed to "Br atom concentrations were"**

Line 170, 'which would also release Br atoms in equal quantities' and 'Here I assume that of the total desired chlorine release.., 20% of that by mass of bromine is released in equal parts $Br_2$ and BrCl'. It is confusing to sometimes consider amount (number of atoms/molecules) and

sometimes massit would perhaps be preferrable to stick with one or the other or to explain carefully. If the mass fraction of bromine to chlorine release is 0.2, then as a mole fraction, only 9% as much Br as Cl is released, as the ratio of the atomic mass of Br to Cl is 2.25. It would be worth mentioning this in the text. Also Hossaini [2016] uses 35% BrCl and 65% $Br_2$. Do you have a reference or an explanation for using equal parts instead?

**We clarify the text as follows: "Here I assume that of the total desired chlorine release (1,250 Tg/yr as in the $Cl_2$-ocean simulation), 20% of that by mass of bromine is released in equal parts $Br_2$ and BrCl (resulting in 1193 Tg/yr $Cl_2$, 187 Tg/yr $Br_2$, and 129 Tg/yr BrCl). This scenario represents a bounding case if artificial sea salt containing bromine impurities were to be continuously emitted as part of an AOE method. In prior work, increasing the flux of natural sea salt aerosol in GEOS-Chem led to relative increases in tropospheric-wide reactive bromine that were comparable to that of reactive chlorine (Horowitz et al., 2020)." (lines 195-200)**

Line 176 change 'dCl2/dt' to $d[Cl_2]/dt$. Note that 'molec' is not a unit and should not be used as a unit, see IUPAC and SI nomenclature references.

**Fixed this and the units in this section.**

Line 206, change 'chloride emissions' to 'chlorine emissions'? Check use of chloride vs. chlorine throughout.

**We add text to clarify throughout this paragraph to distinguish the aerosol chloride and iron emissions, including: "For accumulation mode aerosol chloride emissions of 1,250 Tg (the same mass of total chlorine emissions as the $Cl_2$ experiment), this is 565 Tg/yr pFe." (lines 244-245)**

Table 2. Preferred usage is that if for example $\Delta[H_2O_2]_{high} = -11.1\%$, this equation can be rearranged by dividing both sides by the unit to yield $\Delta[H_2O_2]_{high}/\% = -11.1$. The lhs is used as the label of a column, row or axis, and then the value in the table, or that is plotted in a figure, is a pure number in this case -11.1. The unit is found in the table. It is unconventional to give the unit of some of the values, '%' in the table caption instead of in the table.

**We add % labels throughout.**

Line 243 it could be useful here to explain the mechanism leading to Cl increase, presumably the OH + HCl reaction.

**We add Table S6 below (new numbering) to the SI listing potential reactions involved that are sources of Cl and $Cl_2$. We also add another sentence: "For example, in addition to the reactions in Table S6 which are sources of gas-phase $Cl_y$ rather than cycling between species, OH can react with HCl to produce Cl atom."**

| | | Reaction rate (gas-phase) or reactive uptake coefficient $\gamma$ (heterogeneous) | GEOS-Chem reference |
|---|---|---|---|
| ***OH reactions directly producing Cl and Cl$_2$*** | | | |
| [†]OH + Cl$^-$ | $\rightarrow$ 0.5Cl$_2$ + OH$^-$ | $\gamma = 0.04$[Cl$^-$] | Wang et al. (2019) |
| [†]OH + CH3Cl | $\rightarrow$ Cl + HO2 + H2O | 1.96E-12exp(-1200/T) | Eastham et al. (2014); updated to JPL 15-10 |
| [†]OH + CH2Cl2 | $\rightarrow$ 2Cl + HO2 | 2.61E-12exp(-944/T) | Sherwen et al. (2016) |
| [†]OH + CHCl3 | $\rightarrow$ 3Cl + HO2 | 4.69E-12exp(-1134) | Sherwen et al. (2016) |
| [†]OH + CH3CCl3 | $\rightarrow$ 3Cl + H2O | 1.64E-12exp(-1520/T) | Eastham et al. (2014) |
| OH + HCFCs | $\rightarrow$ $X$Cl + H2O* | Varies | Eastham et al. (2014); updated to JPL 15-10 |
| ***Chlorine reactions producing CO*** | | | |
| CH2O + Cl | $\rightarrow$ CO + HCl + HO2 | 8.1E-11exp(-30/T) | Sherwen et al. (2016) |
| CH3Cl + Cl | $\rightarrow$ CO + 2HCl + HO2 | 2.17E-11exp(-1130/T) | Eastham et al. (2014); Sherwen et al. (2016) |
| CH2Cl2 + Cl | $\rightarrow$ CO + HCl + 2Cl + HO2 | 1.24E-12exp(-1070/T) | Sherwen et al. (2016) |
| CHCl3 + Cl | $\rightarrow$ CO + HCl + 3Cl + HO2 | 3.77E-12exp(-1011/T) | Sherwen et al. (2016) |
| ***Methane oxidation reactions*** | | | |
| CH4 + OH | $\rightarrow$ CH3O2 + H2O | 2.45E-12exp(-1775/T) | JPL 15-10 |
| CH4 + Cl | $\rightarrow$ CH3O2 + HCl | 7.10E-12exp(-1270/T) | JPL 15-10 |

[†]*These reactions have an analogous equivalent involving bromine*

*\*X = 1 for HCFC22 and HCFC142b; X = 2 for HCFC141b and HCFC123.*

Line 252: "the same amount of total chlorine" is misleading, because in the CL2 scenario Cl$_2$ is emitted, while in the other scenarios Cl$^-$ is emitted (one is reactive, the other is not).

**We revise this sentence as follows: "The Cl2, Chloride, and Iron_Chloride experiments have the same amount of total chlorine emissions (see Table 1) but vastly different effectiveness at increasing the Cl atom concentration due to the emitted species (gas-phase Cl2 or particulate chloride) having different reactivities and reactions."**

Line 300 see previous comment on '1-5%, please check, modify to 0.8 top 3.3% as may be indicated.

**Checked and updated as mentioned previously to have a lower limit of 0.23% for consistency.**

line 527, recommend changing 'demand' to 'production'

**Changed**

Line 561 note that the organization is called 'The National Academies of Sciences, Engineering, and Medicine' and is also known as 'The National Academies', but it is not called 'the National Academies of Science'.

**Good catch, "National Academies of Science" corrected to "National Academy of Sciences" for consistency with contract language.**

Line 617 Update reference to Gorham et al. the final paper has been published [Gorham, 2024].

**Updated throughout paper and in references.**

In the Supplemental Information Figure S2, the data formats in the boxes are not standard between the OH and Cl sections of the figure. The blue boxes say for example '1.6x107 Tg/yr' while the green boxes could say 'pFe: 565Tg/yr'. Note that there should always be a space between number and unit, add space to read '565 Tg/yr'. Questions include, mass of what, particles or active iron or iron? Recommend using the same nomenclature throughout e.g. '1.6x107 Tg(H2O2)/yr' and '565 Tg(pFe)/yr'

**Thank you for catching the missing space. I update the figure and the caption for clarity.**

Table S2 give units for the reaction rate coefficients. Remember that 'molecule' is not a unit.

**Fixed**

Table S5, are numbers like 41.2 percentages? It does not say. As noted previously this should be indicated in the heading so for example 'OH_mid / %'. Similar in Table S6 and S7, indicate what are percentages.

**I add % to the column headings.**

Table S6 and S7, there is a stray '40' and '55' in the right column of the respective tables? Maybe this is a line number, but it should not be there.

**The previous table S7 was removed. Numbers have been updated in Table S6 (now Table S9) and on my end there are no stray numbers now.**

**References**

Chen, Q., Wang, X., Fu, X., Li, X., Alexander, B., Peng, X., Wang, W., Xia, M., Tan, Y., Gao, J., Chen, J., Mu, Y., Liu, P., & Wang, T. (2024). Impact of molecular chlorine production from aerosol iron photochemistry on atmospheric oxidative capacity in North China. Environmental Science & Technology, 58(28), 12585–12597. https://doi.org/10.1021/acs.est.4c02534

Gorham, K.A., Abernethy, S., Jones, T.R., Hess, P., Mahowald, N.M., Meidan, D., Johnson, M.S., van Herpen, M.M., Xu, Y., Saiz-Lopez, A. and Röckmann, T., 2024. Opinion: A research roadmap for exploring atmospheric methane removal via iron salt aerosol. Atmospheric Chemistry and Physics, 24(9), pp.5659-5670.

Hossaini, R., Chipperfield, M. P., Saiz-Lopez, A., Fernandez, R., Monks, S., Feng, W., Brauer, P., & Von Glasow, R. (2016). A global model of tropospheric chlorine chemistry: Organic versus inorganic sources and impact on methane oxidation. Journal of Geophysical Research: Atmospheres, 121(23), 14271–14297. https://doi.org/10.1002/2016JD025756

Johnson, M. S., E. J. K. Nilsson, E. A. Svensson, S. Langer, Gas Phase Advanced Oxidation for Effective, Efficient In Situ Control of Pollution, Environmental Science & Technology 48(15), 8768–8776, 2014.

Li, Q., D. Meidan, P. Hess, J. A. Añel, C. A. Cuevas, S. Doney, M. S. Johnson, D. E. Kinnison, R. P. Fernandez, M. van Herpen, L. Höglund-Isaksson, J.-F. Lamarque, T. Röckmann, N. M. Mahowald, A. Saiz-Lopez, Global environmental implications of atmospheric methane removal through chlorine-mediated chemistry-climate interactions, *Nature Communications* **14**(4045), 2023.

Lim, M., Chiang, K., & Amal, R. (2006). Photochemical synthesis of chlorine gas from iron (III) and chloride solution. Journal of Photochemistry and Photobiology A: Chemistry, 183(1-2), 126-132.

Mak, J.E., Kra, G., Sandomenico, T. and Bergamaschi, P., 2003. The seasonally varying isotopic composition of the sources of carbon monoxide at Barbados, West Indies. Journal of Geophysical Research: Atmospheres, 108(D20).

Meidan, D., Li, Q., Cuevas, C. A., Doney, S. C., Fernandez, R. P., van Herpen, M. M., Johnson, M. S., Kinnison, D. E., Li, L., Hamilton, D. S., Saiz-Lopez, A., Hess, P. and N. M. Mahowald (2024). Evaluating the potential of iron-based interventions in methane reduction and climate mitigation. *Environmental Research Letters*, 19(5), 054023.

Meusinger, C., A. B. Bluhme, J. L. Ingemar, A. Feilberg, S. Christiansen, C. Andersen, and M. S. Johnson, Treatment of Reduced Sulphur Compounds and SO2 by Gas Phase Advanced Oxidation, Chemical Engineering Journal, 307, 427-434, 2017.

Mikkelsen, M. K., Liisberg, J. B., van Herpen, M. M. J. W., Mikkelsen, K. V., and Johnson, M. S.: Photocatalytic chloride-to-chlorine conversion by ionic iron in aqueous aerosols: a combined experimental, quantum chemical, and chemical equilibrium model study, Aerosol Research, 2, 31–47, https://doi.org/10.5194/ar-2-31-2024, 2024.

National Academies of Sciences, Engineering, and Medicine. 2024. A Research Agenda Toward Atmospheric Methane Removal. Washington, DC: The National Academies Press. https://doi.org/10.17226/27157.

Pennacchio, L., M. van Herpen, D. Meidan, A. Saiz-Lopez and M. S. Johnson (2024a). Catalytic efficiencies for atmospheric methane removal in the high-chlorine regime. ChemRxiv. 2024; doi:10.26434/chemrxiv-2023-3r8sf-v2

Pennacchio, L., Mikkelsen, M. K., Krogsbøll, M., van Herpen, M., and Johnson, M. S. (2024b). Physical and practical constraints on atmospheric methane removal technologies. Environmental Research Letters, 19(10), 104058.

Röckmann, T., Van Herpen, M.M., Brashear, C., Van der Veen, C., Gromov, S., Li, Q., Saiz-Lopez, A., Meidan, D., Barreto, A., Prats, N., Marmol, I., Ramos, R., Banos, I, Arrieta, J.,

Zaehnle, S., Jordan, A., Moossen, H., Timas, H., Young, D., Sperlich, P., Moss, R. and Johnson, M. S., (2024). The use of δ13C in CO to determine removal of CH4 by Cl radicals in the atmosphere. Environmental Research Letters.

Saiz-Lopez, A., Shillito, J. A., Coe, H., and Plane, J. M. C.: Measurements and modelling of I2, IO, OIO, BrO and NO3 in the mid-latitude marine boundary layer, Atmos. Chem. Phys., 6, 1513–1528, https://doi.org/10.5194/acp-6-1513-2006, 2006.

Sander, R., Keene, W. C., Pszenny, A. A. P., Arimoto, R., Ayers, G. P., Baboukas, E., Cainey, J. M., Crutzen, P. J., Duce, R. A., Hönninger, G., Huebert, B. J., Maenhaut, W., Mihalopoulos, N., Turekian, V. C., and Van Dingenen, R.: Inorganic bromine in the marine boundary layer: a critical review, Atmos. Chem. Phys., 3, 1301–1336, https://doi.org/10.5194/acp-3-1301-2003, 2003.

van Herpen, M. M., Li, Q., Saiz-Lopez, A., Liisberg, J. B., Röckmann, T., Cuevas, C. A., Fernandez, R. P., Mak, J. E., Mahowald, N. M., Hess, P., Meidan, D., Stuut, J.-B., Johnson, M. S., Photocatalytic chlorine atom production on mineral dust–sea spray aerosols over the North Atlantic, Proceedings of the National Academy of Sciences 120(31), e2303974120, 2023.

Wittmer, J., Bleicher, S., Ofner, J., & Zetzsch, C. (2015). Iron(III)-induced activation of chloride from artificial sea-salt aerosol. Environmental Chemistry, 12(4), 461–475. https://doi.org/10.1071/EN14279

Wittmer, J., Bleicher, S., & Zetzsch, C. (2015). Iron(III)-induced activation of chloride and bromide from modeled salt pans. The Journal of Physical Chemistry A, 119(19), 4373–4385. https://doi.org/10.1021/jp508006s

Zhu, X., Prospero, J.M., Savoie, D.L., Millero, F.J., Zika, R.G. and Saltzman, E.S., 1993. Photoreduction of iron (III) in marine mineral aerosol solutions. Journal of Geophysical Research: Atmospheres, 98(D5), pp.9039-9046.

**Citation**: https://doi.org/10.5194/egusphere-2024-3139-RC1